# The dynamics of entrepreneurial agglomeration formation: Social selection and simulation

**Yong Tang, Sohail Ahmad Javeed** [ORCID] *

Management School, Hunan City University, Yiyang, China

* sohailahmaduaf@yahoo.com

**Data Availability Statement:** All data files are attached with Supporting information files.

**Funding:** This paper was supported by National Social Science Fund of China (No. 21BGL199)", but

## Abstract

From the facts that numerous regions with initially similar economic conditions end up with different levels of entrepreneurial agglomeration, this paper constructs a model assuming that the sequential entrants make their career choices based on existing entrepreneurial ratio and describing the dynamics of entrepreneurial agglomeration formation. After mathematical analysis and numerical simulation using NetLogo, it is found that under social selection, a nonlinear Polya process with self-reinforcing and path-dependency characters will emerge, and the repeated entrants' career choices will lead to the agglomeration of entrepreneurship; the agent's risk compensation value, the initial population of agents, the number of role models in the early stage and the initial entrepreneurial ratio are determinants to the formation of entrepreneurial agglomeration. The findings confirm that entrepreneurship has "memory" and the entrepreneurial history could have influence on the future. In order to forge the entrepreneurial agglomeration, our suggestions include exerting influence on the determinants from an early age, and improving the individual's risk-taking abilities.

## 1. Introduction

In order to motivate more entities to sustain China's development, Chinese government adopted the policy of "mass entrepreneurship and innovation" in 2014. Since then, different levels of governments, from top tier to the root tier, have issued numerous policies to increase the entrepreneurial ratio regionally and enhance the entrepreneurial performance for start-ups. After many years of practice, however, the output is not so attractive. Fig 1 exhibits the contrast of several cities in the number of entrepreneurial activity holders to the number of employees for other agencies from 2010 to 2019. Wenzhou, the first city to initiate entrepreneurship since China's Reform and Opening-up started in 1978, still dominate the most entrepreneurial one in China. In contrast, some central and western cities, such as Luohe and Baise, are always at the low level. From the world view, Silicon Valley, Boston, Tel Aviv, Bangalore, et al., are world-class and time-honored leading entrepreneurial agglomerations. However, many more regions, including the regions dotted by many "high-tech parks" or "pioneering parks", cannot forge effective entrepreneurial agglomerations. According to statistics by Hunan Province, China, there are more than 2100 high-tech or pioneering parks in this province, but more than 500 of them accommodate less than 10 start-ups for each one.

the funder had no role in study design, date collection and analysis, decision to publish, or preparation of the manuscript.

**Competing interests:** The authors have declared that no competing interests exist.

In entrepreneurship study, scholars have unanimously confessed the significance of agglomeration of entrepreneurship on regional development, and they always suggested forging more concentrated areas to hold aggregated entrepreneurship due to sustaining economic development [1–3]. Yet from the cases above, we address the problem of high variance in the formation of entrepreneurial agglomeration. As the observations, some regions have been bestowed similar economic and cultural resources, even with equivalent policies to support their entrepreneurship, but their levels of entrepreneurship are highly distinct. Numerous cases show that the prestigious entrepreneurial agglomerations are more time-tested, sustainable, and keeping their soaring entrepreneurial ratios. In contrast, the traditional low-level or the newly-built regions, entrepreneurship cannot be aggregated easily as others.

Entrepreneurial agglomeration is reflected as the aggregated entrepreneurship. Thus, the majority of study on the formation of agglomeration will build the connections between individual's career choices with characteristics of specific agglomeration. As prior studies proved, the new career chooser will definitely make his decision by many criterion, and some internal factors such as his advantages of experience [4, 5], attitudes [6, 7], risk-taking capabilities [8, 9], the propensity in the career type [10, 11], his expectation in the future career [12, 13] etc., or some external factors such as regional opportunities [14], favorable environment [15], or the social status [16], etc.. Though the final career choice is determined by both internal and external factors, prior research have mostly assumed that the individual is well informed before his decision, no matter by his own endeavor or through his social connections or on new business model [17–19].

However, to our knowledge, the existing study may be weak on answering two questions on entrepreneurial agglomeration formation: the first one is why there are different levels of entrepreneurial agglomerations in different regions, even in the regions with similar economic and social conditions. According to our observation, this different level of agglomeration enlightens us to assume that entrepreneurship has "memory" and the entrepreneurial history could have influence on the future. However, prior studies haven't given answers to it. The second is that prior studies have set their research on too strict assumptions, especially when the

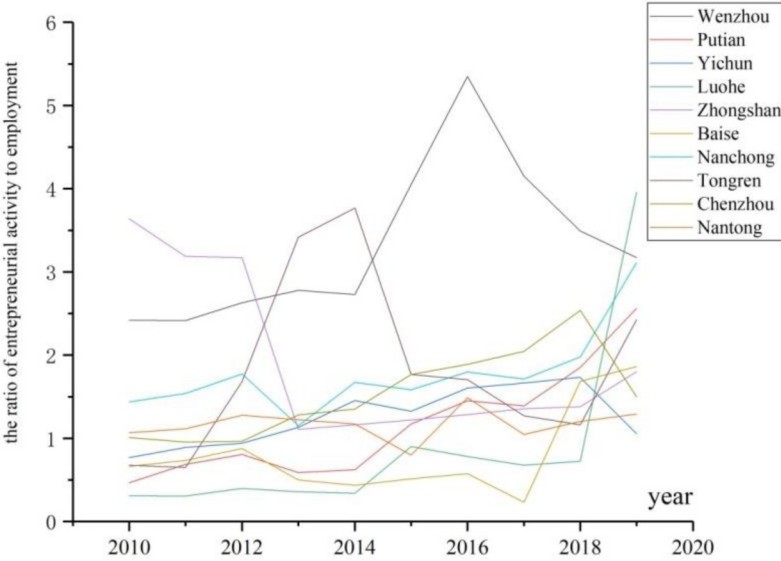

**Fig 1. The level of entrepreneurial activities in several cities of China.**

individual's capabilities on obtaining and utilizing information are considered. As mentioned before, prior literatures have mostly assumed that the individual is well informed before his entrepreneurial decision. However, for most individuals, the region's social circumstances, such as prospect of employment, business environment, education, or the supporting methods, could be barriers to their perception and not unanimously understood or obtained due to exchange cost or other reasons [20]. That is to say, for most people, they cannot wholly understand the situations for the given region. Under this event, the individuals will refer to some easily obtained information as the criteria on career choice.

The level of entrepreneurial agglomeration can be expressed as entrepreneurial ratio for any given region [21], and it can be easily obtained when the region's business facilities or local people are observed [22]. In order to discover the mechanism of entrepreneurial agglomeration formation, we will focus more on the index of entrepreneurial ratio. And, since the entrepreneurial ratio is determined by individual career choices between entrepreneurship and employment, our point will be transferred to the dynamic of micro-decisions of career choices on entrepreneurial agglomeration formation. Our purpose is to provide a theoretical framework describing the dynamics of entrepreneurial agglomeration formation and answer 3 questions: what are the factors that determine the entrepreneurial agglomeration? why dose entrepreneurship mostly prosper in some regions while not in other regions?, and why the region with similar situations cannot replicate the entrepreneurial agglomeration as others?

Unlike traditional research which focused on individual's attributes or on the environmental factors that affect the individual's choice on entrepreneurial career, our assumption is that the new entrant determined to make his career choice cannot fully obtain or understand the region's circumstances. Alternatively, some information or resources, such as the proximity to incumbent entrepreneurs, could be easily obtained or understood. Specifically, if a new entrant can observe the existing ratio of entrepreneurship in the region, his perception about the likelihood to join this career will be reinforced. That is to say, the latter one will refer to the entrepreneurial ratio produced by the precedent's involvement, which is kind of social selection. We assume all the entrants are heterogeneous, and their reactions vary when faced with different level of entrepreneurial activities. Based on social selection and after repeated choices sequentially, the process will converge to some fixed points, and this region will exhibit the landscape of variation when entrepreneurship ratio is concerned. Furthermore, different from prior study that mostly adopt traditional empirical method, we use mathematical method and numerical simulation to depict the repeated entering processes and reveal the dynamics of the level of existing entrepreneurial activity on the formation of entrepreneurial agglomeration. We think the mathematical analysis can better reflect the mechanism of career choices for individuals and explain the determinants for entrepreneurial agglomeration formation, and the numerical simulation can better depict the whole formation process step by step, which are always regarded as "black box" for traditional empirical studies [23].

This study makes two major contributions. First, we advocate that the formation of entrepreneurial agglomeration is a process of social selection by individuals. Specifically, we found that the career choice of individuals is affected by social environment, especially by the observable conditions such as the scale of existing entrepreneurship. And after repeated selection by the individuals, the agglomeration will emerge following the rule of Polya process. Not like most studies that confess the new agents will make their decision on entrepreneurship based on clear information, out study insists that the individuals can make their choices based on very basic information, and the existing entrepreneurship will have a self-reinforcing capability to absorb or expel the new entrant. This finding confirms that entrepreneurship has "memory" and the entrepreneurial history could have influence on the future. Second, along with our study, we confirm that some determinants from social selection can affect the

formation of entrepreneurial agglomeration. In detail, our results indicate that the agent's risk compensation value, the initial scale of agents, the number of entrepreneur role models from the region's early stage and the initial entrepreneurial ratio in the region can be involved to contribute the formation of entrepreneurial agglomeration. Thus exerting influence on the determinants from an early age, and improving the individual's risk-taking abilities are beneficial to the formation of entrepreneurial agglomeration.

This paper is organized as follows: in Section 1, we introduce the background, some concepts and the motivation to conduct this study. In Section 2, we review the literatures and build the basic model. Section 3 is to conduct mathematical analysis of the entrepreneurial agglomeration formation process. And section 4 is the simulation and discussion. The last section is the conclusion and suggestions.

## 2. Literature review and the basic model

### 2.1 Literature review

In economics studies, agglomeration, one phenomenon or platform to cluster resources or entities, has been widely discussed for a long time. For its capability in collecting factors and optimizing resource distribution [24, 25], absorbing and disseminating knowledge [26], facilitating the market structures and reducing exchange costs [27, 28], some effects from agglomeration has been summarized as "agglomeration theory" [29–31]. Agglomeration has been deeply and widely accepted in social and economic circles, in either real or virtual form. There are many synonymous forms that have been accepted in academic and industrial circles. The typical ones include geographical or economic convergence [32, 33], clusters [34], aggregations [35], concentrations [36], or even eco-system for some industrial areas [37]. In order to express the clustering attributes, the gauge of agglomeration is always on the ratio of activities for the given region [21], the ratio of related investment for the given region [38], or the areas covered by related activities [39].

Along with the tendency of entrepreneurship being stressed, entrepreneurial agglomeration, the form of start-ups and entrepreneurs concentrating geographically, has been intensively paid attention to in recent years. Accordingly, the discussion of its formation process, as well as the factors affecting the formation, is highlighted. As for the exploring the mechanism to entrepreneurial agglomeration formation, existing literatures can be categorized into 3 branches, and most research are conducted empirically:

The first one is the examination of personal attributes on the formation of entrepreneurial agglomeration. This theory argues the agglomeration formation is the result of repeated choices by self-sponsored agents, and the agents will evaluate his personal attributes before joining in some area's entrepreneurship. As discussed by Kihlstrom and Laffont [40], the choice of entrepreneurship and employment is determined by individual's risk-averse propensity, and the agents who are more risk-taking will choose entrepreneurship. When more risk-taking agents join in, the region's entrepreneurial activities will increase and the entrepreneurial agglomeration takes shape [9]. Also, some other attributes such as educational background [41], the preference to be more independent and self-control [42, 43], be more connected with acquaintance through social network [44, 45], and be more self-employed [46], are all included in the discussion. It is worth noting that though most literatures supporting this theory confess the personal attributes are the major determinants for entrepreneurial agglomeration, some other factors, such as the region's conditions are more or less involved as well [14–16].

The second one is the "economic advantage" on the formation of entrepreneurial agglomeration. This theory stresses the importance of local economic conditions. From the observations of some time-honored entrepreneurial agglomeration, it is easy to identify numerous

advantages to benefit entrepreneurship. And this theory has the assumption that the individuals can basically recognize and utilize the economic situations in the region. According to Mei, et al. [47], in China's rural areas, the most successful entrepreneurial agglomerations appear in the regions dominated by natural resource such as charming landscape or precious primary goods. The similar finding has been proved by Bas and Kunc [48], who confirms in less developed countries, the entrepreneurial agglomerations will appear firstly at the resource-abundant sites, such as mine areas. Also, the lately produced resources, such as convenient infrastructure facilities [49], the government-sponsored facilities such as incubators [50], or favorable policies supporting entrepreneurship [51, 52], etc., can also benefit the formation of entrepreneurial agglomeration.

The third one is on the social advantage. Social advantage stresses the social norm, social interaction, social preference and sociocultural factors [53]. He, et al. [54] confirmed that if an area is more advocating on entrepreneurship, such as honoring more entrepreneurs and entrepreneurial activities, this area will be more likely to attract entrepreneurship. Also, the entrepreneurial culture and history are effective factors for the formation and consolidation of the entrepreneurial agglomeration. Audretsch, et al. [55] argued the importance of identity cognition on the creating new businesses. In their research, the individuals under entrepreneurial culture are more acceptable to the identity of entrepreneurship and easily absorbed to take similar career. Furthermore, the social resources can facilitate the formation of entrepreneurial agglomeration. Zheng and Du [38] points out that if a person can easily meet the role models in entrepreneurship, he may welcome this career and make him one of them with higher probability. Andersson and Larsson [56] stressed the importance of social interaction, and verified that individuals' decisions to become entrepreneurs are influenced by entrepreneurial neighbors. Besides, in a well-built agglomeration, the new entrants who have entrepreneurial intentions will be more easily credited and internally involved by the incumbent entrepreneurs [37, 57, 58]. It is worth noting that the research above had all assumed that the individuals who are facing career choice are clear about the social laws and the interaction rules.

Prior literatures have provided abundant evidences on the formation of agglomeration. As can be seen, most scholars admit the agglomeration is a process of repeated career choices by potential entrepreneurs, and they have also drawn up some factors that affect the formation of entrepreneurial agglomeration. However, it cannot explain the situation aforementioned that some regions are always dominating the entrepreneurial roles while other regions keep stagnant even with long time endeavor, and the region with similar situations cannot replicate the entrepreneurial agglomeration as others. As for the deficiencies, the main points include two: the first is on the assumption that the individuals can rationally understand all information that the region owns, so they can always make the right choice to enter into and form an agglomeration; and the second is that prior research is mostly rooted on empirical study based on static data, which always build the connections between the new entrants and existing entrepreneurial agglomeration, while ignoring the trajectory of agglomeration formation process. Theoretically, if the individuals can make use of the economic and social advantages fully under the assumption of information transparency or effects of path-dependency [59], entrepreneurship will always be reinforced and the increasing return to entrepreneurship will hold on continuously [60], which will result in the region's long-time sticky to all potential entrepreneurs and finally keep on growing in its agglomeration. However, our cases in the first section have demonstrated that this argument cannot be supported. Practically, for the majority of new entrants, they are not explicitly knowledgeable to the landscape of the region. That is to say, they cannot capture all "economic advantages" or "social advantages" by themselves. Even though they understand the goodness to be the entrepreneurs and their attributes are suitable

to entrepreneurship, they cannot be supportive to make the career choices when information is ambiguous, much less they can match their attributes with the features in some specific region.

Therefore, our assumption is that the new entrant will seek some easily obtained information or resources to decide his career. Specifically, in any region, if all agents are randomly distributed, the existing ratio of entrepreneurship is often expressed as his chance to meet entrepreneurs, and this information is much easier to acquire and be understood [21], so we keep the ratio of entrepreneurship as a very important variable in our analysis. Also, we adopt the variable of risk-taking ability as suggested by Putniņš and Sauka [9], because the new entrant can fully evaluate his affordability if he is determined to start a new business. And we think traditional empirical study cannot depict the whole formation process step by step, we will make a trial by using mathematical analysis and numerical simulation to compensate the weakness.

## 2.2 Basic model

In social studies, bandwagon effect is a very important concept and a very important social selection mechanism [61]. When individuals cannot capture all information on making their choices, they will transfer to the social selection, i.e., the surrounding choices made by their equivalents because they want to be part of the majority, just like in a bandwagon parade for politics, people voting for the candidate who appears to have the most support. Bandwagon effect on the entrepreneurial choice has been proved by many scholars [62, 63]. In their remarks, entrepreneurship could be regarded as an increasing-return process with respect to adoption, which can be evidenced by high entrepreneurship areas where large concentration will definitely lower the ambiguity for taking entrepreneurship and reinforce the entrepreneurial activities. In other words, if the new entrant can meet more incumbent entrepreneurs, their willingness to take the same choice will be reinforced. Therefore, the number of entrepreneurship or the entrepreneurial ratio is an important variable in determining individual career choices. Also, because the local number or the ratio of entrepreneurship will change along with the increase or decrease of new entrepreneurial activities, the formation of entrepreneurial agglomeration can be seen as a dynamic process of self-reinforced adoption by individuals. Our basic model that reflects this social selection process is as Fig 2.

As shown in Fig 2, we set initial entrepreneurial ratio (ER in Fig 2), and suppose the new entrant 1 (the potential entrepreneur 1) makes his choice based on the ratio. If he is determined to take entrepreneurship, the ER in the region will be increased, as shown in the upper branch; or he can choose to be employed, which will bring the ER down, as shown in the lower branch. This repeated chooses will be conducted by entrants 2, 3, . . ., $i$, $j$. As a result, the process will show the character of self-reinforcing and positive feedback along the trajectory, which can also exhibit the formation of agglomeration dynamically. Section 3 will detail this dynamic process using mathematical analysis.

# 3. The mathematical analysis of the entrepreneurial agglomeration formation process

## 3.1 The social selection criterion of entrepreneur and employment

Consider a region where income is obtained by entrepreneurship or by employment.

The agent's cost is a quadratic function in the work supplied (labeled as $L$), and the wage is labeled as $w$. So the agent's cost is $\sigma_0 + \sigma_1 L + \sigma_2 L^2$, where $\sigma_0$, $\sigma_1$, and $\sigma_2$ represent the coefficients of agent's attributes such as education, family background, capital, etc., and the income is $wL$.

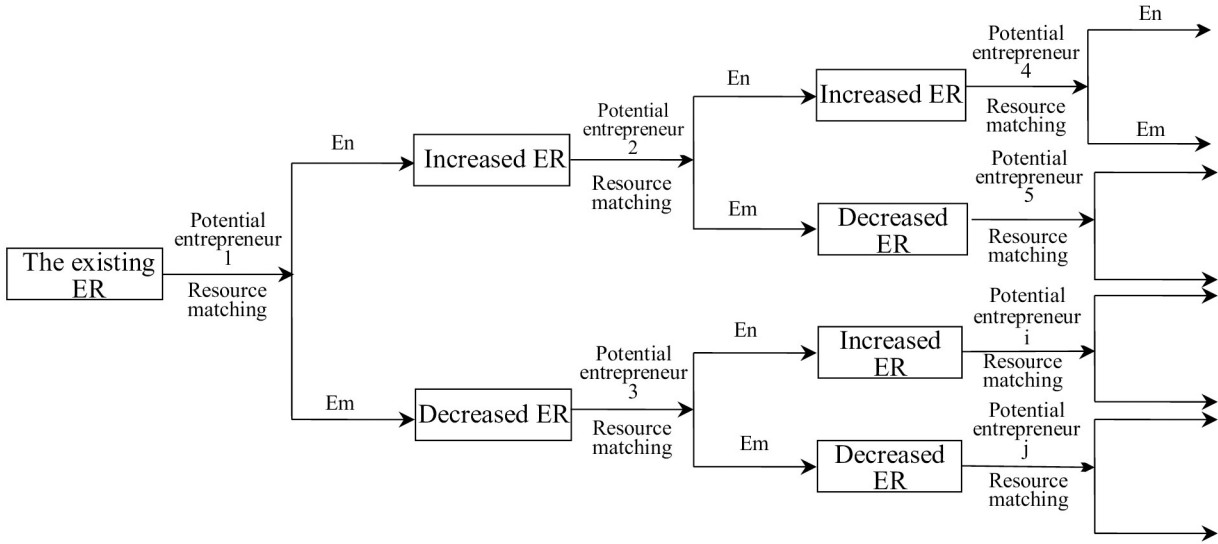

**Fig 2. The social selection of entrepreneurial agglomeration formation.**

Therefore, for any agent $i$, the net return has the form:

$$r^i = w^i L - (\sigma_0^i + \sigma_1^i L + \sigma_2^i L^2) \tag{1}$$

Set $\max(r^i)$, each agent will supply an optimal amount of work to guarantee the equal margin of income and cost if and only if:

$$w^i = \sigma_1^i + 2\sigma_2^i L \tag{2}$$

So, we have $L^{i*} = (w^i - \sigma_1^i)/(2\sigma_2^i)$, indicating the optimal work for either entrepreneurship or employment.

Substituting $L^{i*}$ into $r^i$, we have:

$$r^{i*} = w^i(w^i - \sigma_1^i)/(2\sigma_2^i) - \sigma_0^i - (\sigma_1^i(w^i - \sigma_1^i)/(2\sigma_2^i) - (\sigma_2^i(w^i - \sigma_1^i)^2)/(4(\sigma_2^i)^2) \tag{3}$$

Rewrite (3) as:

$$r^{i*} = -\sigma_0^i + (w^i - \sigma_1^i)^2)/(4(\sigma_2^i)^2) \tag{4}$$

Because the Eq (4) summarizes the net return for entrepreneurship or employment, we separately use subscript $h$ and $e$ to reflect the situations for two different kinds of careers, and omit the superscript to reflect all agents. Thus,

$$r_h^* = -\sigma_{0h} + (w_h - \sigma_{1h})^2)/(4\sigma_{2h}) \text{ and } r_e^* = -\sigma_{0e} + (w_e - \sigma_{1e})^2)/(4\sigma_{2e}) \tag{5}$$

For any given region, we set the initial entrepreneurial ratio $p$. Besides, let $\sigma_{ih} = (1 + p)\gamma_{ie}$, yields:

$$r_e^* = ((w_e - \gamma_1(1 + p))^2/(4(1 + p)\gamma_2) - (1 + p)\gamma_0 \tag{6}$$

Under the condition that no risk for entrepreneurship, the agent will make the career choice by whether $r_e^* - r_h^* > 0$ or $r_e^* - r_h^* < 0$. If all agents are risk-neutral, the one who prefers to be an entrepreneur must consider the risk circumstances in the region. Accordingly, they

need a higher net return to compensate the risk when taking entrepreneurship. Set $\theta^j$ as the risk compensation value for entrepreneurship. Apparently, $\theta^j$ is not fixed because when the region has more information to make decision, the agent will ask for lower level of compensation. So, we express the risk compensation value for any agent $j$ as:

$$\theta^j/(1+p) \tag{7}$$

When risk compensation value is incorporated, the criterion for entrepreneurship choice is expressed as:

$$re_e^{*j} - re_h^{*j} > \theta^j/(1+p) \tag{8}$$

The absolute net return for entrepreneurship to employment is

$$c^j = -\theta^j + (1+p)(re_e - re_h) \tag{9}$$

Now consider in a given region, the relationship between entrepreneurship and employment is highly impacted by each other. From one hand, the employment rate will be increased when more agents choose entrepreneurship; from another hand, more entrepreneurship will incur more labor demands, as well as higher paying for workers. Thus we express the labor equilibrium as:

$$\int L(w_h, p)dp = 1 - p \tag{10}$$

Set the wage for workers as $w_h = \varepsilon_0 + \varepsilon_1 p$ and income for entrepreneurship as $w_e = \omega_0 + \omega_1 p$.

Substituting expressions of (2)–(8) and the wage rates into Eq (9), we obtain formula (11):

$$c^j = a_0^j + a_1 p + a_2 p^2 + a_3 p^3 \tag{11}$$

Where:

$$a_0^j = -\theta^j + \sigma_0 + (\gamma_0 \varpi_0^2 + \gamma_0 \gamma_1^2 - 2\varpi_0 \gamma_0^2 - 2\gamma_1 \varpi_0)/(4\gamma_2) - (\varepsilon_0^2 + \sigma_1^2 - 2\varepsilon_0 \sigma_1)/(4\sigma_2)$$

$$a_1 = \sigma_2 - 2\gamma_0 + (2\gamma_1^2 - 2\omega_0 \omega_1 \gamma_0 - 2\gamma_0 \gamma_1 \omega_1)/(4\gamma_2) + (4\varepsilon_0 \varepsilon_1 + 2\varepsilon_1 \sigma_1 - \varepsilon_0^2 - \sigma_1^2)/(4\sigma_2)$$

$$a_2 = -\gamma_0 + (\gamma_0 \omega_1^2 + 2\gamma_1 \omega_1 + \gamma_1^2)/(4\gamma_2) - (\varepsilon_1^2 + 2\varepsilon_0 \varepsilon_1 - 2\varepsilon_1 \sigma_1)/(4\sigma_2)$$

$$a_3 = -\varepsilon_1^2/(4\sigma_2)$$

Apparently, the agent chooses entrepreneurship if and only if $c^j > 0$.

For the reason that all agents are heterogeneous, $c^j$ is not only determined by the agents' characteristics (such as $-\theta^j$, $\sigma_i$), but also affected by social factors such as the entrepreneurial ratio ($p \in [0, 1]$) and the coefficients of $\gamma_i$, $\varepsilon_i$, $\omega_i$ in the region. Also, because all parameters are determined by individuals or social environment, and any agent, $a_0^j$, $a_1$, $a_2$ and $a_3$ are constant, so all of them can be considered as the coefficients for determining the entrepreneurial choice.

Assuming a region has a given number of agents. Although heterogeneous in attributes, the individuals will similarly respond to the changes of entrepreneurial ratio. Reflected in the equation, each agent's relative net return can be expressed as the vertical displacement from a common function $f$ [62].

In turn, (11) can be rewritten in the form of $f(p)$ as:

$$c^j = a_0^j + f(p) \tag{12}$$

Where $f(p) = a_1 p + a_2 p^2 + a_3 p^3$.

Ultimately, the Eq (12) is the social selection criterion for entrepreneurship and employment because this criterion is rooted in the previous whole ratios. It could be increasing or decreasing to entrepreneurial ratio $p$.

## 3.2 The formation process of entrepreneurial agglomeration

In this section, we will explore the trajectory of entrepreneurial agglomeration from its start to its final situation.

Back to $c^j = a_0^j + f(p)$. When $p = 0$, $c^j = a_0^j$. Thus, the vertical displacement is determined by $a_0^j$. Let the region be formed by a continuum of agent uniformly distributed along the closed interval $[a_0^0, a_0^1]$. Because all agents are heterogeneous, so $a_0^j$ is specific and $a_0^0 < a_0^j < a_0^1$. Connecting to the reality, even in the most favorable situation, not all agents will simultaneously choose entrepreneurship, so we set $a_0^0 < 0$.

We also need to construct the probability function $g(p)$ to reflect a new agent's likelihood to choose entrepreneurship. As aforementioned uniformly distributed agents, every agent faces the criterion of $c^j$, and the probability function $g(p)$ can be expressed as:

$$g(p) = (a_0^1 + f(p))/((a_0^1 + f(p)) - (a_0^0 + f(p))) = (a_0^1 + f(p))/(a_0^1 - a_0^0) \tag{13}$$

Eq (13) describes the interdependence among agents' decisions, in which each agent in influenced by what other agents have chosen before him, or the probability of next agent choosing entrepreneurship is a function of the probability that prior agents are entrepreneurs at the current entrepreneurial ratio.

We then detail this process:

As in Fig 2, let there be an initial $N$ of agents in the given region (both entrepreneurs and workers are included), and suppose there are $E_0$ of entrepreneurs, thus the initial ER is $p_0 = E_0/N$.

Suppose there will be agents sequentially enter into this region. The first agent determined to make the career choice is labeled as "1". So when he chooses entrepreneurship, the ER will change to be $p_1 = (E_0 + 1)/(N + 1)$; and when he chooses to be employed, the ER will be $p_1 = E_0/(N + 1)$. Now move to the right side of Fig 2. Suppose there are $n$ new entrants, the region will have $N + n$ agents in total. We set the ER as $E_n$. When one more agent (the $(n+1)th$ agent) enters in, the amount of agents is $N + n + 1$. We set $E_{n+1}$ as its ER. We express $p_n$ and $p_{n+1}$ as:

$$p_n = E_n/(N + n) \tag{14}$$

$$p_{n+1} = E_{n+1}/(N + n + 1) \tag{15}$$

If the $(n+1)th$ agent chooses entrepreneurship, or when $g(p_n)$ happens, we have:

$$p_{n+1} = (p_n(N + n) + 1)/(N + n + 1) = p_n(1 - 1/(N + n + 1)) + 1/(N + n + 1) \tag{16}$$

And when he chooses employment, or when $1 - g(p_n)$ happens, we have:

$$p_{n+1} = p_n(N + n))/(N + n + 1) = p_n - 1/(N + n + 1) \tag{17}$$

The expectation of $p_{n+1}$ is calculated in the form of (18):

$$Exp(p_{n+1}) = (p_n(1 - 1/(N + n+)) + 1/(N + n+))g(p) + (p_n - 1/(N + n + 1))(1 - g(p))$$
$$= p_n(N + n)/(N + n + 1) + g(p)/(N + n + 1) \tag{18}$$

The conversion formula of (18) is:

$$Exp(p_{n+1}) - p_n = (g(p) - p_n)/(N + n + 1) \tag{19}$$

Then, define the function $\phi(p)$. Let $\phi(p) = 1$ when new entrant becomes entrepreneur and let $\phi(p) = 0$ when new agent chooses to be employed. Apparently, $\phi(p)$ is a sign function. Relating to the definition of expectation value, the combination of (16) and (17) is as:

$$p_{n+1} = p_n + (g(p) - p_n)/(N + n + 1) + (\phi(p) - g(p))/(N + n + 1) \tag{20}$$

Eq (20) depicts the relation between $p_n$ and $p_{n+1}$, or the ER for the entrant $n$ and the ER for entrant $n + 1$. For $p_{n+1}$, it is determined by two parts: the first is $p_n + (g(p) - p_n)/(N + n + 1)$, the deterministic part of $p$; and second is $(\phi(p) - g(p))/(N + n)$, the stochastic part of $p$. According to the definition, we have $Exp((\phi(p) - g(p))/(N + n + 1)) = 0$. Then, we can predict the whole process will converge with probability 100% to the fixed point of $g(p)$. That relation between $p_n$ and $p_{n+1}$ is a typical positive feedback, in which $p_n$ can contribute deterministic and stochastic parts to $p_{n+1}$, and this process conforms to a nonlinear Polya process [64, 65].

According to rule of Polya process, when $g(p) > p_n$, the probability of the next agent choosing entrepreneurship is higher than the entrepreneurship ratio, so the expectation value of $p_{n+1}$ is bigger than $p_n$; correspondingly, when $g(p) < p_n$, the next new agent will have a higher probability to be employed and we will have $Exp(p_{n+1}) < p_n$. With this repetition, $p$ will gradually converge to a fixed value expressed as $g(p)$, and the fixed $g(p)$ is final entrepreneurial agglomeration.

Next, we move to the fixed points which reflect entrepreneurial agglomeration. In light with intermediate value theorem: for different $n$, $g(p)$ could be bigger or less than $p_n$, and $g(p)$ and $p_n$ are continuous, then there will exist one or more fixed $p^*$ that satisfies the equation of $g(p^*) = p^*$. Alternatively, the solution to $p^*$ is to solve the equation group:

$$\begin{cases} g(p) = (a_0^1 + f(p))/(a_0^1 - a_0^0) \\ f(p) = a_1^j p + a_2^j p^2 + a_3^j p^3 \\ g(p) = p_n \end{cases} \tag{21}$$

Equation group (21) can be transformed to (22):

$$(a_0^1 + a_1^j p + a_2^j p^2 + a_3^j p^3)/(a_0^1 - a_0^0) = p \tag{22}$$

As for Eq (22), the coefficients $a_1$, $a_2$ and $a_3$ are functions of $\varepsilon_0, \varepsilon_1, \omega_0, \omega_1, \gamma_0, \gamma_1, \sigma_0, \sigma_1$ and therefore are exogenous. The interval of $[a_0^0, a_0^1]$ is uniformly distributed agents, so it is also exogenous. The risk compensation value $\theta^j$ is indigenous to new agent, and it only has relation with $a_0^j$. In sum, if new entrants repetitively make their career choices, the entrants' risk compensation value $\theta^j$ and related ER $p_j$ will contribute a lot when the region converges to some fixed points in its level of entrepreneurship.

## 4. Simulation and discussion

### 4.1 Simulation

This section will demonstrate the trajectory of entrepreneurial agglomeration formation by numerical simulation, so as to explore the determinants that pose impact on this trajectory.

Firstly, we calculate the fixed points for a given region with detailed numerical information. As for Eq (22), it can be divided into two functions: $f_1(p) = (a_0^1 + a_1^j p + a_2^j p^2 + a_3^j p^3)/(a_0^1 - a_0^0)$ and $f_2(p) = p$. In $f_1(p)$ and $f_2(p)$, the coefficients $a_1, a_2, a_3$ and $a_0^j$ are functions of parameters of $\varepsilon_0, \varepsilon_1, \omega_0, \omega_1, \gamma_0, \gamma_1, \sigma_0, \sigma_1$ and $\sigma_2$. According to the rule of over identification for equations, theoretically we can directly assign values to $a_1, a_2, a_3$ and $a_0^j$. However, when $f_1(p)$ and $f_2(p)$ are practically analyzed, the value assignment will be confined to some intervals: from one hand, when the entrepreneurial ratio increases, the newly entrant will reduce their ambiguity for entrepreneurship risks, so their risk compensation value of $\theta^j$ will be lowered; from another hand, high entrepreneurial ratio will incur high demand of employees, and the salary paid by the entrepreneurs will be improved. So, $f(p)$ is an S-shape curve in the interval of $p \in [0,1]$. Accordingly, $g(p)$ is a linear conversion of $f(p)$, thus $g(p)$ is also an S-shape curve. For the reasons above, we set $a_1 = 0, a_2 = 4.5, a_3 = -3, a_0^j \in [-1.6, 0.02]$, and the plot of graphs of $f_1(p)$ and $f_2(p)$ is as Fig 3. (when $p \in [0,1]$).

The solutions of $p$ are:

$$p_1 = 0.0124, p_2 = 0.5643, p_3 = 0.9281$$

Which are the possible final entrepreneurial ratios for a given region.

In order to verify the solutions, we then simulate the repeated choices by new entrants and observe the trajectory of convergence in entrepreneurial ratios. We adopt multi-agent simulation software NetLogo to simulate the processes.

Step 1: initiation. Set initial population $N = 50$, the initial ER is $p = 33.3\%$ (the red agents are workers, and the green agents are entrepreneurs, and all agents are randomly distributed on the patch (as shown in Fig 4).

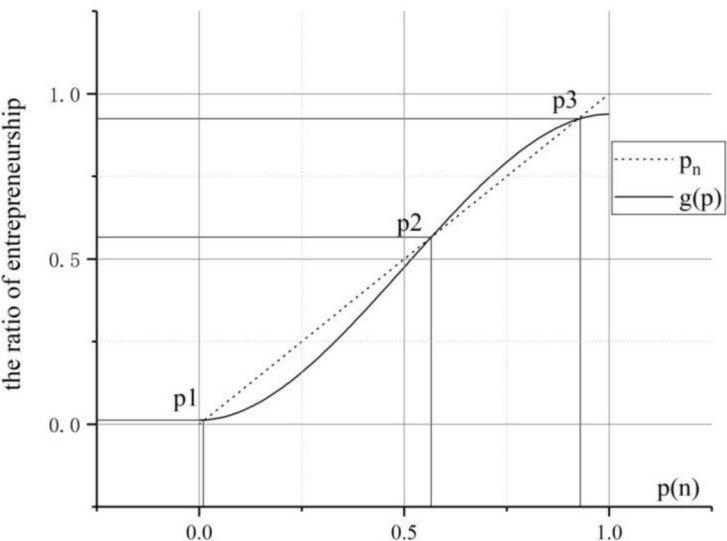

**Fig 3. The exploration of fixed points to entrepreneurial agglomeration.**

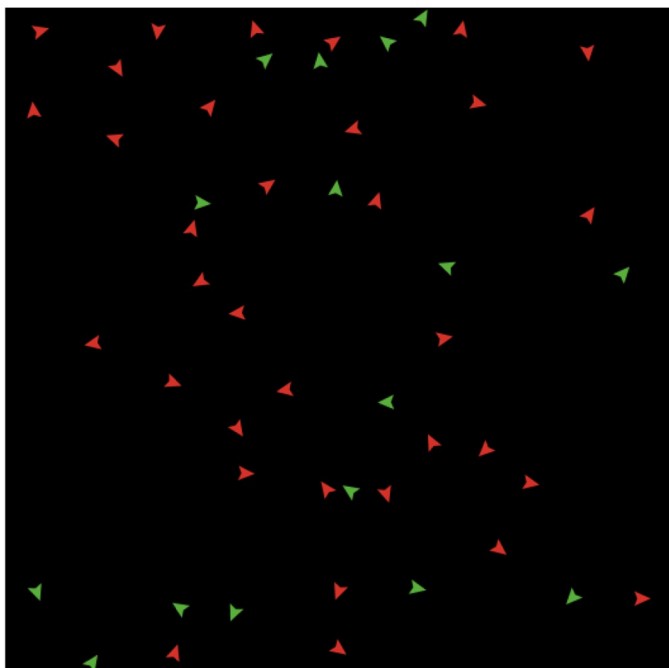

**Fig 4. The initial agents (N = 50).**

Step 2: parameter setting. As the numerical example, we set $a_1 = 0$, $a_2 = 4.5$ and $a_3 = -3$, and $a_0^j \in [-1.6, 0.02]$. In NetLogo, we use the code "(random-float 1 < (4.5 * (ratio ^ 2) − 3 * (ratio ^ 3)" to reflect the probability.

Step 3: one-time simulation. In order to make comparison, we set $n = 300$, $n = 1000$ and $n = 10000$, which means there will be 300, 1000 and 10000 agents entering into this given region one by one. By one time simulation, the graphs are as Fig 5.

As shown in Fig 5, through one-time simulation, the red agents (or workers) are dominating the patches, showing more and more agents will choose employment in the next choices. Particularly, in the graph of $n = 10000$, there are very small number of entrepreneurs, indicating that the repeated choices have overrun the entrepreneurial choices, and this region faces worsen situation to form the entrepreneurial agglomeration.

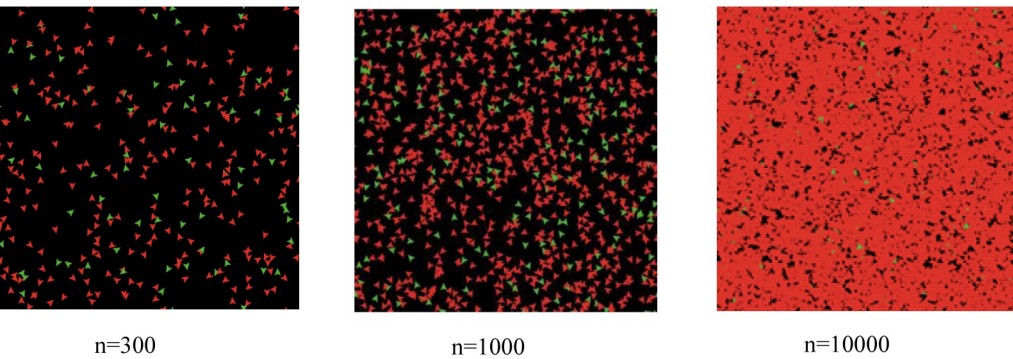

n=300                            n=1000                            n=10000

**Fig 5. The one-time simulation.**

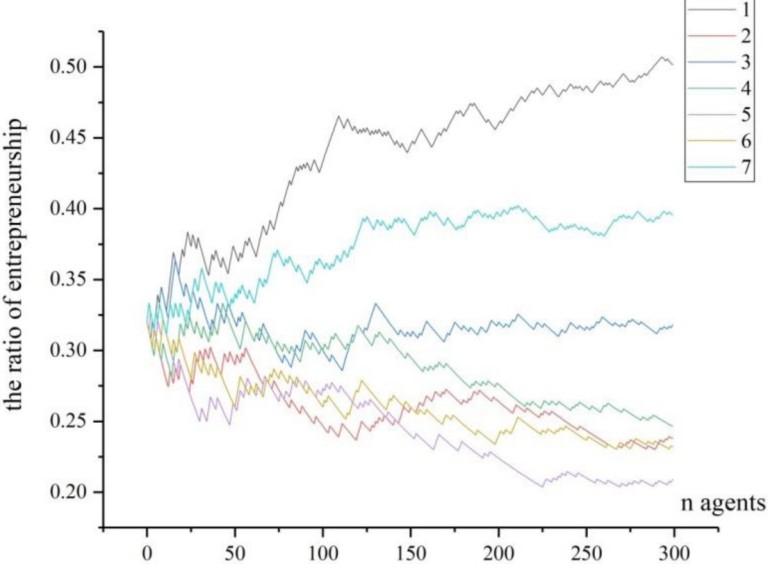

**Fig 6. The trajectories of 7 times of simulation (n = 300).**

Step 4: the arrival of fixed ER points. We move back to test the fixed points calculated in Fig 3. In order to reflect the trajectory that the ER converges to the fixed point, we record all agents' choices when $n = 300$, $n = 1000$ and $n = 10000$, and conduct 7-time of simulation for each situation. All entrepreneurial ratios are recorded by Figs 6–8.

From Fig 6, there are 300 agents entering into this region, and from 7 times of simulation, we can see various entrepreneurial ratios appear for different simulations. Noticing that the number of agents is not big, some curved lines are intertwined to each other when observed

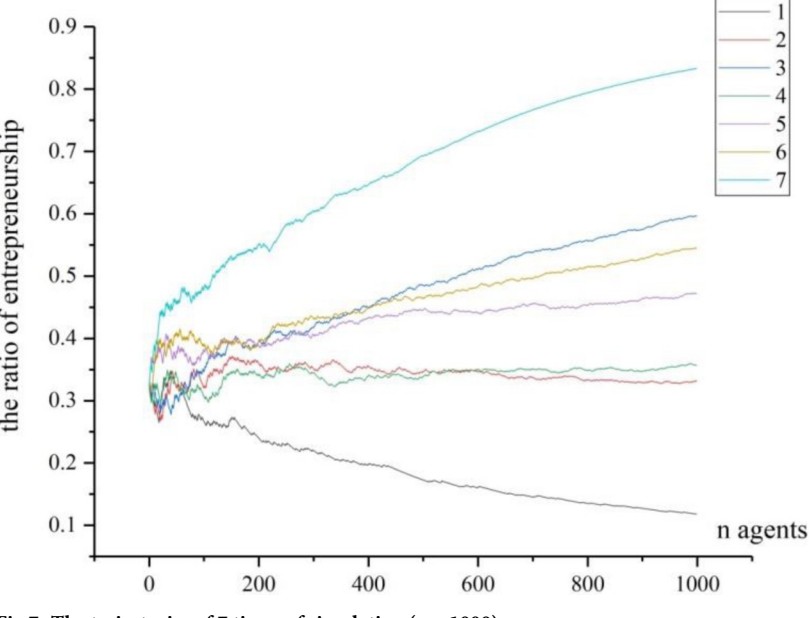

**Fig 7. The trajectories of 7 times of simulation (n = 1000).**

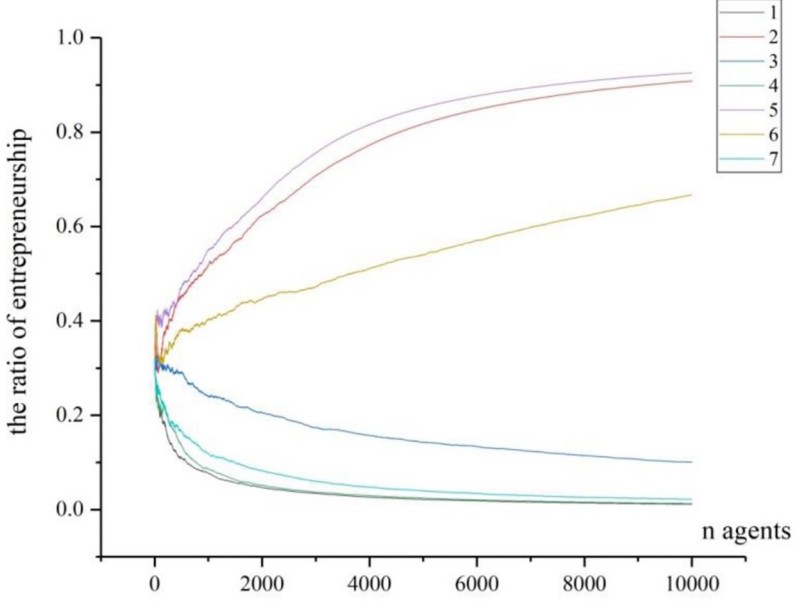

**Fig 8. The trajectories of 7 times of simulation (n = 10000).**

from the start to the end, indicating that the entrepreneurial ratios for each simulation are not stable. Thus, in the short run, the region cannot clearly present stable entrepreneurial agglomeration.

From Fig 7, when 1000 agents enter into this region one by one, the curves from 7 times of simulation are more separated to each other, signifying the tendency to convergence appears. Now that the number of agents is not very big, we cannot judge whether the curves will lead to the fixed points calculated in Fig 3.

And from Fig 8, there are over 10000 agents, which is a huge number compared with previous simulation. Also, 7 times of simulation are conducted, and the curves are much smoother than the other two situations. With the increase of agents entering into the region, the variation appears significantly. For the processes of number 1 and number 2, the curves move upward gradually and up to the similar high point of above 0.9, approaching to the fixed point of 0.9281. For the processes of number 3, 4 and 7, the curves move downward gradually and down to the similar point of below 0.1, almost approaching to the fixed point of 0.0124. And for the process of number 6, the curve between the others moves upward a little but apparently keeps stable to a middle level of entrepreneurial ratio. From the scale we can basically observe that it moves to above 0.5, which is nearly approaching to the fix point of 0.5643.

Step 5: resetting initial entrepreneurial ratio.

In Figs 6 to 8, though the ER for each simulation could stay at different level, the probability for them to stay at low level is bigger, because we set the initial ER at 33.3%, much less than 50%. We then change the initial ER to 0.1, 0.2, 0.3, 0.4, 0.5, 0.6, 0.7, 0.8 and 0.9, and we set the number of agents as 100000. After the simulations, we have all the graphs in Fig 9.

From the observations of Fig 9, we can see that the smaller initial entrepreneurial ratios (such as 10%, 20%, 30% etc.) are basically converged to a lower final ER (as shown in the figure, most curves are approaching to almost lowest points of 0.0124). Conversely, for the bigger initial entrepreneurial ratios (such as 60%, 70%, 80% and 90%), their curves are promptly approaching to the high fixed point of 0.9218 unanimously. From the results above, we find

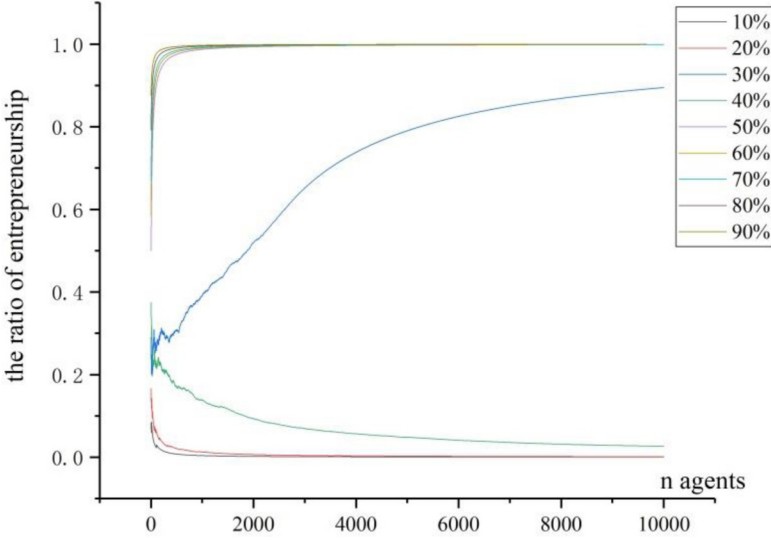

**Fig 9. The trajectories under different initial ratios.**

that basically, the level of initial ER has influence on the final result, and bigger (smaller) initial value will lead to bigger (smaller) final value with high probability. But there are also some exceptions. Taking the curve signed by 40% as an example, our simulation result shows that this initial ER leads to a higher level of final entrepreneurial ratio, very contrary to the smaller initial value that leads to smaller final value. And for the curve arising from 50% of initial entrepreneurial ratio, the converging point is approaching to the highest point, which has not conformed to the rule summarized above. So, our conclusion is that the initial entrepreneurial ratio can partly determine the final result, but the converging process is very complicated, some other factors can also contribute to the formation of agglomeration.

## 4.2 Discussion

Based on the model in section 3 and the simulation results in section 4.1, we further our discussion on the determinants for the entrepreneurial agglomeration.

(1) The individual's risk compensation value on the formation of entrepreneurial agglomeration

Unlike the study by Kihlstrom and Laffont [40], we argue that the agents are all risk-neutral but their risk-taking behaviors can be compensated by premiums. As in the Eq (11), smaller $\theta^j$ will lead to bigger $a_0^j$, so as to move $f_1(p)$ along with $p$ vertically, resulting in upward moving of the fixed values of entrepreneurial ratio. The result has some theoretical and practical implications: Theoretically, for any agent, if he has low level of risk compensation, the entrepreneurial ratio will increase. In other words, the strong risk-taking ability for the individuals could stimulate higher entrepreneurial agglomeration, and the lower risks for taking entrepreneurship will also lower the value of risk compensation value and lead to higher entrepreneurial agglomeration too. Practically, because the strong risk-taking ability is the propensity from the specific agent, so the methods such as entrepreneurial education, entrepreneurial experience can contribute to it [66]; Furthermore, the reduction of the potential risks from the environment can also lower the risk compensation value for individuals, so possible practical methods include creating fair businesses environment,

supporting more resources to the individuals, or nurturing more business opportunities et al [67].

(2) The initial scale of agents on the formation of entrepreneurial agglomeration
From the comparison of Figs 6–8, we can conclude theoretically that the initial scale of agents can greatly affect the entrepreneurial ratio. To our knowledge, this conclusion has seldom been mentioned by scholars [38, 47, 54]. As to our explanation, when the agents are in small number (see Fig 6), one entrepreneur in this region may have high impact on the entrepreneurial ratio, thus it can greatly affect the next agent's choice, and the region's entrepreneurial ratio will vary dramatically. With the increasing of agents in the region, one particular entrepreneur cannot significantly affect the final entrepreneurial ratio. As a result, individual's contribution to the final result is not significant. From the global perspective, when the agents in the region are in great number, the final entrepreneurial ratio is not determined by separated choice, but by the social selection arising from individuals. Our explanation of social selection is from the wholeness of entrepreneurship in the region, by which the individuals will reinforce themselves to follow some tracks which can be predicted under some condition. The social selection has typical scale effect. When the number in the social group is small, the circumstance cannot form its special characters as in Fig 6. Under this condition, even after repeated choices by the new agents, the given region cannot present significant entrepreneurial ratio. However, when more and more agents join in, the given region will contain great number of information for the agents to make their choices. And each agent's choice contributes to the ultimate aggregation, though not with equal contribution. Consequently, the final results of entrepreneurial ratios will be consistently converged to some fixed points. Our result partly confirms the theory provided by Andersson and Larsson [56], who emphasized the importance of entrepreneurial neighbors on the individuals' decisions to become entrepreneurs. The practical implication of this conclusion is that for the region where entrepreneurial agglomeration is advocated, it should contain a certain number of agents (the potential entrepreneurs or workers) at its early age. In another word, the population of labors for the region is the basis for entrepreneurial agglomeration formation, especially in the early age.

(3) The role model in the region on the formation of entrepreneurial agglomeration to Minniti, et al. [68], the imitation of entrepreneurial activity across different markets encourages more individuals to become entrepreneurs. Our model theoretically proved that the new agents will imitate the agents who have become incumbent entrepreneurs, and the role models of entrepreneurship are very essential to the formation of entrepreneurial agglomeration, but they will present a decreasing capability in stimulating entrepreneurship. This conclusion basically supports the theory by Zheng and Du [38] and Minniti, Bygrave and practice [68]. While in our study, the function of role model is a little different from theirs. Our assumptions are that the individuals will evaluate the ER and then decide whether they can accept the risks of entrepreneurship in the region. A role model is more successful in creating new jobs and starting new firms, thus he can provide more information and spillover more knowledge to the surroundings. When new entrants can observe more role models and get more information about entrepreneurship for a given region, their risk-taking abilities will accordingly be improved and their sense to start new businesses will be encouraged. However, as shown in our simulation, the role models will present a decreasing capability in stimulating entrepreneurship. The possible explanation could be that when the number of agents becomes huge, the small number of role models cannot compete with the whole situation, as shown in section 5.1; furthermore, the role model will dominate the market and absorb more employment, which could threaten the new entrepreneurs and

potentially reduce the entrepreneurial activities. From practical perspective, this conclusion implies that the introducing more role models early is more effective in concentrating the entrepreneurial activities.

(4) The initial ER in the region on the formation of entrepreneurial agglomeration

Through mathematical analysis and numerical simulation we have basically proved that when the initial ER in a region is higher, the probability to converge at a high level will increase (as shown in Fig 9). This theoretical conclusion is conformed to the bandwagon effect on the entrepreneurial choice [62, 63, 69], which implies that the entrepreneurial history of the region is important. When a region is with higher initial ER, the individuals will face high probability to meet the entrepreneurs, and they will also be easily absorbed in to take similar career. Under such situation, the region will exhibit the character of path dependency. Another reason is that high initial ER will clear away more ambiguity for entrepreneurship, with which the new entrants can be more inclined to this kind of career. This conclusion partially confirmed the "social advantage" theories [55–57] and "economic advantage" theory [51], but our conclusion emphasizes the increasing-return process to adoption, and this process is dynamic and fluctuating before it comes to the final agglomeration, which could be a new finding to our knowledge. For the reason that the initial ER in the region is more important on the formation of entrepreneurial agglomeration that later ER, the practical meaning is that introducing more entrepreneurial activities or increasing the initial ER when the region is at small scale or in early stage is more effective in forming the agglomeration.

## 5. Conclusion and suggestion

### 5.1 Conclusion

The entrepreneurial agglomeration, as an effective means to sustain regional economic development, has been widely emphasized globally, especially in emerging countries such as China. However, numerous cases show that the fostering and consolidating entrepreneurial agglomeration is not an easy task. For lots of regions with similar social and economic circumstances, the ultimate results for the entrepreneurial agglomeration vary dramatically though the local governments have paid great endeavors in forging them. Then we are facing the problems of "what are the factors that determine the entrepreneurial agglomeration", "why entrepreneurship mostly prospers in some regions while not in other regions?" and "why the region with similar situations cannot replicate the entrepreneurial agglomeration as others?"

In order to explore the determinants of entrepreneurial agglomeration formation, traditional literatures have mostly sought evidences from the individuals or the regional conditions. However, as our observation, the individuals cannot clearly obtain and understand all necessary information and resources for entrepreneurship in the given region. That is to say, traditional information for the new entrants is generally ambiguous. Along with this judgement, the individuals will seek for much easier criterion to make their career choices. Entrepreneurial agglomeration is reflected as the aggregated entrepreneurship. For any region, the new entrant is fully exposed to the environment where he is with some probability to meet the entrepreneurs, and the bandwagon effect will drive him much easily affected by his proximity to take similar activity. Under such argument, we provide a theoretical framework describing the dynamics of entrepreneurial agglomeration formation. Unlike the traditional research that assume the individuals are informed all the necessary information in the given region, we introduce the hypothesis that the new entrants will make their career choices (entrepreneurship or employment) based on existing entrepreneurial ratio. According to our model, after repeated choices, this trajectory conforms to a nonlinear Polya process, i.e., the whole process

will emerge the self-reinforcing and path-dependency characters. Then after numerical calculation and simulation, we confirmed that the process under our assumption will converge to some fixed points, which can exhibit the formation of entrepreneurial agglomeration and explain why different regions will vary in their levels of entrepreneurial agglomeration.

Our main findings are concentrated on the discovery of the dynamic process of and the determinants for the formation of entrepreneurial agglomeration. Under our assumption, our model can explain the formation of entrepreneurial agglomeration is a nonlinear Polya process, and the repeated entrants' career choices will lead to the convergence of entrepreneurship in any given region. As to the determinants, our analysis and simulation have proved at least four categories can be included: the agent's risk compensation value, the initial scale of agents, the number of role models (especially in the early stage), and the initial entrepreneurial ratio. Accordingly, they can basically answer three questions in section 1.

## 5.2 Implications

(1) Practical implications

Firstly, considering the strong effects from "memory" and "history" in entrepreneurship, the policy should be more concentrated on the determinants from "early" age, such as attracting more role models and introducing more entrepreneurial activities when the region is at its initiation stage or in small scale. In detail, the government can start with the evaluation of a region's entrepreneurial activities and make up polices to gather more role entrepreneurs and potential entrepreneurs into the region by cutting off taxes, providing more subsidies, or issuing more favorable measures to start and operate the businesses. Secondly, due to the great contribution on self-reinforcement effect, lowering the risk compensation value for individuals is very important. The policy point can focus on fostering the traits on entrepreneurship. In detail, the government should invest more on improving entrepreneurship education, encouraging more potential entrepreneurs to observe, witness or participate in the incumbent business to enhance their entrepreneurial experience, so as to improve the individuals' risk-taking abilities

Finally, for the reason that the improvement of entrepreneurial infrastructure is a good means to lower the entrepreneurship risks, and accordingly reduce the risk compensation value for the potential entrepreneurs. The policy can exert its ability on creating a fairer businesses environment, building more entrepreneurial facilities such as incubators, transportation and communication systems etc., to nurture more business opportunities and facilitate more entrepreneurial activities.

(2) Theoretical implications

Firstly, we explore the dynamics of entrepreneurial formation. It is proved that the formation of entrepreneurial agglomeration is a self-reinforcing process. Based on social selection by individuals, we confirm that the formation of entrepreneurial agglomeration can be described as a Polya process, and the existing entrepreneurship will absorb or expel the new entrant. In our theoretical framework, the entrepreneurship has "memory" and the entrepreneurial history could have influence on the future, which can be expressed as another kind of bandwagon effect emerging from entrepreneurial activities.

Secondly, we explore the determinants which contribute to the entrepreneurial formation. Based on mathematical analysis and simulation, we conclude that some determinants from social selection can affect the formation of entrepreneurial agglomeration. In detail, our results theoretically indicate that 4 determinants from agents and regions, i.e., the agent's risk compensation value, the initial scale of agents, the number of role models (especially in the early

stage), and the initial entrepreneurial ratio, can contribute to the formation of entrepreneurial agglomeration, thus focusing on improving individual's risk-taking abilities, increasing the initial population scale and entrepreneurial ratio in the region, and introducing more entrepreneur role models in the early age can improve the formation of entrepreneurial agglomeration.

### 5.3 Limitations and direction for future research

One limitation of our study is confined to theoretical analysis, because robustness, validation and accuracy of our conclusions are not verified empirically. Our study adopts the model proceeded by mathematical analysis and numerical simulation, which can clearly exhibit the formation process of entrepreneurial agglomeration and conclude the determinants under strict model construction conditions, but have some weakness in introducing the real data to verify whether our conclusions can sufficiently reflect the real situations. Another limitation is that the determinants from our study are only confined to 4 categories; therefore, some other variables from prior research cannot be verified in our model. Our model has strong theoretical foundation, and most adopted determinants are suggested or mentioned by prior study, thus some of our findings also confirm to prior studies. However, limited by the mathematical model, we only adopt some of the variables; whether other variables could be regarded as the dynamics has not been proofed in our analysis.

Future research should modify the mathematical model, making it more acceptable to the variables suggested by prior research and observed from real situations, and more capable in explaining the mechanism of entrepreneurial agglomeration formation. In addition, empirical analysis combining with our theoretical model and using real data from various regions is also recommended in the future study.

## Supporting information

**S1 File.**
(RAR)

## Acknowledgments

The authors would like to acknowledge the support provided by Management School, Hunan City University, Yiyang, China.

## Author Contributions

**Conceptualization:** Yong Tang.

**Data curation:** Yong Tang.

**Formal analysis:** Yong Tang.

**Funding acquisition:** Yong Tang.

**Investigation:** Yong Tang.

**Methodology:** Yong Tang.

**Supervision:** Sohail Ahmad Javeed.

**Validation:** Sohail Ahmad Javeed.

**Writing – review & editing:** Sohail Ahmad Javeed.

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
