## [Decision Letter · Decision Letter 0]

1 Feb 2023

PONE-D-22-27945The Dynamics of Entrepreneurial Agglomeration Formation: Social Selection and SimulationPLOS ONE

Dear Dr. Javeed,

Thank you for submitting your manuscript to PLOS ONE. After careful consideration, we feel that it has merit but does not fully meet PLOS ONE’s publication criteria as it currently stands. Therefore, we invite you to submit a revised version of the manuscript that addresses the points raised during the review process.

ACADEMIC EDITOR:  Please take into account the recommendations made by the two reviewers. I believe that all the requested changes are pertinent and must be addressed seriously. 

We look forward to receiving your revised manuscript.

Kind regards,

Valentina Diana Rusu, PhD

Academic Editor

PLOS ONE

and https://journals.plos.org/plosone/s/file?id=ba62/PLOSOne_formatting_sample_title_authors_affiliations.pdf.

“YT

National Social Science Fund of China (No. 21BGL199).

Yong Tang”

3. Thank you for stating the following in the Acknowledgments/ Funding Section of your manuscript:

“This paper was supported by National Social Science Fund of China (No. 21BGL199).”

“YT

National Social Science Fund of China (No. 21BGL199).

Yong Tang”

“No”

Additional Editor Comments:

The reviewers requested changes to the article, so my recommendation is Major revisions, and all requested changes must be taken into account.

Reviewers' comments:

Reviewer's Responses to Questions

**Comments to the Author**

1. Is the manuscript technically sound, and do the data support the conclusions?

Reviewer #1: Partly

Reviewer #2: Yes

2. Has the statistical analysis been performed appropriately and rigorously? 

Reviewer #1: I Don't Know

Reviewer #2: Yes

3. Have the authors made all data underlying the findings in their manuscript fully available?

Reviewer #1: Yes

Reviewer #2: Yes

4. Is the manuscript presented in an intelligible fashion and written in standard English?

Reviewer #1: Yes

Reviewer #2: No

5. Review Comments to the Author

Reviewer #1: This paper is related to an interesting topic, namely The Dynamics of Entrepreneurial Agglomeration Formation: Social Selection and Simulation for PLOS ONE

However, the manuscript needs to be further developed in order to meet the expected academic requirements.

The contribution not clear

In introduction Provide a stronger motivation for the study.

Introduction must include research gap, and research questions, that drive the paper, should be built in the introduction from an ongoing and pertinent bibliography (up to 2023).

Literature Review: the authors need to perform a detailed literature review. Much more critical analysis needs to be undertaken of the relevant literature and key points/issues identified which will underpin the rest of the manuscript. Try to include some latest journal article references published from 2020-2023 in the specific areas to improve the relevance and currency of the research.

Theoretical and practical implications is missing. Provide a more critical discussion regarding theoretical and practical implications. What are the theoretical and practical implications that can be derived from this study?

Reviewer #2: This is an interesting paper and I enjoyed reading it. However, there are essential weaknesses that need to be addressed.

1) The introductory/opening section should communicate a little clearer the literature gaps, as well as the study's aims & objectives in order to facilitate the flow of the study.

2) Overall there are good arguments and well researched points made in this paper, but I feel that author needs to take to a further level. I recommend that you include the following references focused on the target journal and on the paper’s topics:

Al-Mamary, Y. H., & Alshallaqi, M. (2022). Impact of autonomy, innovativeness, risk-taking, proactiveness, and competitive aggressiveness on students’ intention to start a new venture. Journal of Innovation & Knowledge, 7(4), 100239. https://10.1016/j.jik.2022.100239

Bouncken, R. B., Lapidus, A., & Qui, Y. (2022). Organizational sustainability identity:‘New Work’of home offices and coworking spaces as facilitators. Sustainable Technology and Entrepreneurship, 1(2), 100011. https://doi.org/10.1016/j.stae.2022.100011

Dana, L. P., Salamzadeh, A., Hadizadeh, M., Heydari, G., & Shamsoddin, S. (2022). Urban Entrepreneurship and Sustainable Businesses in Smart Cities: Exploring the Role of Digital Technologies. Sustainable Technology and Entrepreneurship, 100016. 1 (2) https://doi.org/10.1016/j.stae.2022.100016

Dijkstra, H., van Beukering, P., & Brouwer, R. (2022). Marine plastic entrepreneurship; Exploring drivers, barriers and value creation in the blue economy. Sustainable Technology and Entrepreneurship, 1(3), 100018. https://doi.org/10.1016/j.stae.2022.100018

Guo, H., Guo, A., & Ma, H. (2022). Inside the black box: How business model innovation contributes to digital start-up performance. Journal of Innovation & Knowledge, 7(2), 100188. https://https://doi.org/10.1016/j.jik.2022.100188

Huang, Y., Li, P., Wang, J., & Li, K. (2022). Innovativeness and entrepreneurial performance of female entrepreneurs. Journal of Innovation & Knowledge, 7(4), 100257. https://10.1016/j.jik.2022.100257

Méndez-Picazo, M., Galindo-Martín, M., & Castaño-Martínez, M. (2021). Effects of sociocultural and economic factors on social entrepreneurship and sustainable development. Journal of Innovation & Knowledge, 6(2), 69-77. https://10.1016/j.jik.2020.06.001

Metallo, C., Agrifoglio, R., Briganti, P., Mercurio, L., & Ferrara, M. (2021). Entrepreneurial Behaviour and New Venture Creation: the Psychoanalytic Perspective. Journal of Innovation & Knowledge, 6(1), 35-42. https://10.1016/j.jik.2020.02.001

3) The research is well-developed.

4) At the end of, the author should include clear statements as to where research should now go.

5) Carefully check the references, so as to make sure they are all complete and follow the Guidelines to Authors.

6) Finally, when you submit the corrected version, please do check thoroughly, in order to avoid grammar, syntax or structure/presentation flaws.

Thank you for the opportunity to read the paper.

6. PLOS authors have the option to publish the peer review history of their article (what does this mean?). If published, this will include your full peer review and any attached files.

Reviewer #1: No

Reviewer #2: No

---

## [Author Response · Author response to Decision Letter 0]

1 Jun 2023

Responses to Reviewer-1

Reviewer #1: This paper is related to an interesting topic, namely The Dynamics of Entrepreneurial Agglomeration Formation: Social Selection and Simulation for PLOS ONE

However, the manuscript needs to be further developed in order to meet the expected academic requirements.

1. The contribution not clear

Response: First of all, we would like to thank reviewer for such valuable suggestions. Your suggestions really helped us to improve this paper. We have added the contribution marked with red color, under section 1 (page 5).

“This study makes two major contributions. First, we advocate that the formation of entrepreneurial agglomeration is a process of social selection by individuals. Specifically, we found that the career choice of individuals is affected by social environment, especially by the observable conditions such as the scale of existing entrepreneurship. And after repeated selection by the individuals, the agglomeration will emerge following the rule of Polya process. Not like most studies that confess the new agents will make their decision on entrepreneurship based on clear information, out study insists that the individuals can make their choices based on very basic information, and the existing entrepreneurship will have a self-reinforcing capability to absorb or expel the new entrant. This finding confirms that entrepreneurship has “memory” and the entrepreneurial history could have influence on the future. Second, along with our study, we confirm that some determinants from social selection can affect the formation of entrepreneurial agglomeration. In detail, our results indicate that the agent’s risk compensation value, the initial scale of agents, the number of entrepreneur role models from the region’s early stage and the initial entrepreneurial ratio in the region can be involved to contribute the formation of entrepreneurial agglomeration. Thus exerting influence on the determinants from an early age, and improving the individual’s risk-taking abilities are beneficial to the formation of entrepreneurial agglomeration.”

2. In introduction Provide a stronger motivation for the study.

Response: As your comment, we refine our paper and add the motivation for the study, all marked in red color, in section 1 (page 4).

“The level of entrepreneurial agglomeration can be expressed as entrepreneurial ratio for any given region, and it can be easily obtained when the region’s business facilities or local people are observed. In order to discover the mechanism of entrepreneurial agglomeration formation, we will focus more on the index of entrepreneurial ratio. And, since the entrepreneurial ratio is determined by individual career choices between entrepreneurship and employment, our point will be transferred to the dynamic of micro-decisions of career choices on entrepreneurial agglomeration formation. Our purpose is to provide a theoretical framework describing the dynamics of entrepreneurial agglomeration formation and answer 3 questions: what are the factors that determine the entrepreneurial agglomeration? why dose entrepreneurship mostly prosper in some regions while not in other regions?, and why the region with similar situations cannot replicate the entrepreneurial agglomeration as others?”

3. Introduction must include research gap, and research questions, that drive the paper, should be built in the introduction from an ongoing and pertinent bibliography (up to 2023).

Response: Thank you for your comments. As your suggestion, we rewrite some parts of our Introduction, and include research gap, research questions, that drive the paper, mostly being built from bibliography. We also include some latest papers as our citation (see section 1 “Introduction”, from page 3 to page 5, all marked with red color).

“As prior studies proved, the new career chooser will definitely make his decision by many criterion, and some internal factors such as his advantages of experience [4,5], attitudes [6,7], risk-taking capabilities [8,9], the propensity in the career type [10,11], his expectation in the future career [12,13] etc., or some external factors such as regional opportunities [14], favorable environment [15], or the social status [16], etc.. Though the final career choice is determined by both internal and external factors, prior research have mostly assumed that the individual is well informed before his decision, no matter by his own endeavor or through his social connections or on new business model [17-19].”

“However, to our knowledge, the existing study may be weak on answering two questions on entrepreneurial agglomeration formation: the first one is why there are different levels of entrepreneurial agglomerations in different regions, even in the regions with similar economic and social conditions. According to our observation, this different level of agglomeration enlightens us to assume that entrepreneurship has “memory” and the entrepreneurial history could have influence on the future. However, prior studies haven’t given answers to it. The second is that prior studies have set their research on too strict assumptions, especially when the individual’s capabilities on obtaining and utilizing information are considered. As mentioned before, prior literatures have mostly assumed that the individual is well informed before his entrepreneurial decision. However, for most individuals, the region’s social circumstances, such as prospect of employment, business environment, education, or the supporting methods, could be barriers to their perception and not unanimously understood or obtained due to exchange cost or other reasons [20]. That is to say, for most people, they cannot wholly understand the situations for the given region. Under this event, the individuals will refer to some easily obtained information as the criteria on career choice.”

“Unlike traditional research which focused on individual’s attributes or on the environmental factors that affect the individual’s choice on entrepreneurial career, our assumption is that the new entrant determined to make his career choice cannot fully obtain or understand the region’s circumstances. Alternatively, some information or resources, such as the proximity to incumbent entrepreneurs, could be easily obtained or understood. Specifically, if a new entrant can observe the existing ratio of entrepreneurship in the region, his perception about the likelihood to join this career will be reinforced. That is to say, the latter one will refer to the entrepreneurial ratio produced by the precedent’s involvement, which is kind of social selection. We assume all the entrants are heterogeneous, and their reactions vary when faced with different level of entrepreneurial activities. Based on social selection and after repeated choices sequentially, the process will converge to some fixed points, and this region will exhibit the landscape of variation when entrepreneurship ratio is concerned. Furthermore, different from prior study that mostly adopt traditional empirical method, we use mathematical method and numerical simulation to depict the repeated entering processes and reveal the dynamics of the level of existing entrepreneurial activity on the formation of entrepreneurial agglomeration. We think the mathematical analysis can better reflect the mechanism of career choices for individuals and explain the determinants for entrepreneurial agglomeration formation, and the numerical simulation can better depict the whole formation process step by step, which are always regarded as “black box” for traditional empirical studies [23].”

4. Literature Review: the authors need to perform a detailed literature review. Much more critical analysis needs to be undertaken of the relevant literature and key points/issues identified which will underpin the rest of the manuscript. Try to include some latest journal article references published from 2020-2023 in the specific areas to improve the relevance and currency of the research.

Response: As your comments, we add more information in the Literature Review, including the logical of literatures and the topics in a more detailed way. We also make much more critical analysis on the Literature Reviews so as to make them more relevant and be more suitable to the coming analysis. Some 2020-2023 literatures are also included in our modification. The major modifications are as below (see sub-section 2.1 “Literature Review”, page 6-9, mainly in red font).

“In order to express the clustering attributes, the gauge of agglomeration is always on the ratio of activities for the given region [21], the ratio of related investment for the given region [38], or the areas covered by related activities [39].

Along with the tendency of entrepreneurship being stressed, entrepreneurial agglomeration, the form of start-ups and entrepreneurs concentrating geographically, has been intensively paid attention to in recent years. Accordingly, the discussion of its formation process, as well as the factors affecting the formation, is highlighted. As for the exploring the mechanism to entrepreneurial agglomeration formation, existing literatures can be categorized into 3 branches, and most research are conducted empirically:

The first one is the examination of personal attributes on the formation of entrepreneurial agglomeration. This theory argues the agglomeration formation is the result of repeated choices by self-sponsored agents, and the agents will evaluate his personal attributes before joining in some area’s entrepreneurship. As discussed by Kihlstrom and Laffont [40], the choice of entrepreneurship and employment is determined by individual’s risk-averse propensity, and the agents who are more risk-taking will choose entrepreneurship. When more risk-taking agents join in, the region’s entrepreneurial activities will increase and the entrepreneurial agglomeration takes shape [9]. Also, some other attributes such as educational background [41], the preference to be more independent and self-control [42,43], be more connected with acquaintance through social network [44,45], and be more self-employed [46], are all included in the discussion. It is worth noting that though most literatures supporting this theory confess the personal attributes are the major determinants for entrepreneurial agglomeration, some other factors, such as the region’s conditions are more or less involved as well [14-16].

The second one is the “economic advantage” on the formation of entrepreneurial agglomeration. This theory stresses the importance of local economic conditions. From the observations of some time-honored entrepreneurial agglomeration, it is easy to identify numerous advantages to benefit entrepreneurship. And this theory has the assumption that the individuals can basically recognize and utilize the economic situations in the region. According to Mei, et al. [47], in China’s rural areas, the most successful entrepreneurial agglomerations appear in the regions dominated by natural resource such as charming landscape or precious primary goods. The similar finding has been proved by Bas and Kunc [48], who confirms in less developed countries, the entrepreneurial agglomerations will appear firstly at the resource-abundant sites, such as mine areas. Also, the lately produced resources, such as convenient infrastructure facilities [49], the government-sponsored facilities such as incubators [50], or favorable policies supporting entrepreneurship [51,52], etc., can also benefit the formation of entrepreneurial agglomeration.

The third one is on the social advantage. Social advantage stresses the social norm, social interaction, social preference and sociocultural factors [53]. He, et al. [54] confirmed that if an area is more advocating on entrepreneurship, such as honoring more entrepreneurs and entrepreneurial activities, this area will be more likely to attract entrepreneurship. Also, the entrepreneurial culture and history are effective factors for the formation and consolidation of the entrepreneurial agglomeration. Audretsch, et al. [55] argued the importance of identity cognition on the creating new businesses. In their research, the individuals under entrepreneurial culture are more acceptable to the identity of entrepreneurship and easily absorbed to take similar career. Furthermore, the social resources can facilitate the formation of entrepreneurial agglomeration. Zheng and Du [38] points out that if a person can easily meet the role models in entrepreneurship, he may welcome this career and make him one of them with higher probability. Andersson and Larsson [56] stressed the importance of social interaction, and verified that individuals’ decisions to become entrepreneurs are influenced by entrepreneurial neighbors. Besides, in a well-built agglomeration, the new entrants who have entrepreneurial intentions will be more easily credited and internally involved by the incumbent entrepreneurs [37,57,58]. It is worth noting that the research above had all assumed that the individuals who are facing career choice are clear about the social laws and the interaction rules.

Prior literatures have provided abundant evidences on the formation of agglomeration. As can be seen, most scholars admit the agglomeration is a process of repeated career choices by potential entrepreneurs, and they have also drawn up some factors that affect the formation of entrepreneurial agglomeration. However, it cannot explain the situation aforementioned that some regions are always dominating the entrepreneurial roles while other regions keep stagnant even with long time endeavor, and the region with similar situations cannot replicate the entrepreneurial agglomeration as others. As for the deficiencies, the main points include two: the first is on the assumption that the individuals can rationally understand all information that the region owns,, so they can always make the right choice to enter into and form an agglomeration; and the second is that prior research is mostly rooted on empirical study based on static data, which always build the connections between the new entrants and existing entrepreneurial agglomeration, while ignoring the trajectory of agglomeration formation process. Theoretically, if the individuals can make use of the economic and social advantages fully under the assumption of information transparency or effects of path-dependency [59], entrepreneurship will always be reinforced and the increasing return to entrepreneurship will hold on continuously [60], which will result in the region’s long-time sticky to all potential entrepreneurs and finally keep on growing in its agglomeration. However, our cases in the first section have demonstrated that this argument cannot be supported. Practically, for the majority of new entrants, they are not explicitly knowledgeable to the landscape of the region. That is to say, they cannot capture all “economic advantages” or “social advantages” by themselves. Even though they understand the goodness to be the entrepreneurs and their attributes are suitable to entrepreneurship, they cannot be supportive to make the career choices when information is ambiguous, much less they can match their attributes with the features in some specific region. 

Therefore, our assumption is that the new entrant will seek some easily obtained information or resources to decide his career. Specifically, in any region, if all agents are randomly distributed, the existing ratio of entrepreneurship is often expressed as his chance to meet entrepreneurs, and this information is much easier to acquire and be understood [21], so we keep the ratio of entrepreneurship as a very important variable in our analysis. Also, we adopt the variable of risk-taking ability as suggested by Putniņš and Sauka [9], because the new entrant can fully evaluate his affordability if he is determined to start a new business. And we think traditional empirical study cannot depict the whole formation process step by step, we will make a trial by using mathematical analysis and numerical simulation to compensate the weakness.”

5. Theoretical and practical implications is missing. Provide a more critical discussion regarding theoretical and practical implications. What are the theoretical and practical implications that can be derived from this study?

Response: Thank you for your comments. We have rewritten our discussion and some theoretical and practical implication as well. The modified contents is as below :

1. In sub-section “4.2 Discussion” (from page 22-25), we add theoretical and practical implication to each finding:

“(1) The individual’s risk compensation value on the formation of entrepreneurial agglomeration

Unlike the study by Kihlstrom and Laffont [40], we argue that the agents are all risk-neutral but their risk-taking behaviors can be compensated by premiums. As in the equation (11), smaller will lead to bigger , so as to move along with vertically, resulting in upward moving of the fixed values of entrepreneurial ratio. The result has some theoretical and practical implications: Theoretically, for any agent, if he has low level of risk compensation, the entrepreneurial ratio will increase. In other words, the strong risk-taking ability for the individuals could stimulate higher entrepreneurial agglomeration, and the lower risks for taking entrepreneurship will also lower the value of risk compensation value and lead to higher entrepreneurial agglomeration too. Practically, because the strong risk-taking ability is the propensity from the specific agent, so the methods such as entrepreneurial education, entrepreneurial experience can contribute to it [66]; Furthermore, the reduction of the potential risks from the environment can also lower the risk compensation value for individuals, so possible practical methods include creating fair businesses environment, supporting more resources to the individuals, or nurturing more business opportunities et al [67]. 

(2) The initial scale of agents on the formation of entrepreneurial agglomeration

From the comparison of Fig. 6, 7 and 8, we can conclude theoretically that the initial scale of agents can greatly affect the entrepreneurial ratio. To our knowledge, this conclusion has seldom been mentioned by scholars [38,47,54]. As to our explanation, when the agents are in small number (see Fig. 6), one entrepreneur in this region may have high impact on the entrepreneurial ratio, thus it can greatly affect the next agent’s choice, and the region’s entrepreneurial ratio will vary dramatically. With the increasing of agents in the region, one particular entrepreneur cannot significantly affect the final entrepreneurial ratio. As a result, individual’s contribution to the final result is not significant. From the global perspective, when the agents in the region are in great number, the final entrepreneurial ratio is not determined by separated choice, but by the social selection arising from individuals. Our explanation of social selection is from the wholeness of entrepreneurship in the region, by which the individuals will reinforce themselves to follow some tracks which can be predicted under some condition. The social selection has typical scale effect. When the number in the social group is small, the circumstance cannot form its special characters as in Fig. 6. Under this condition, even after repeated choices by the new agents, the given region cannot present significant entrepreneurial ratio. However, when more and more agents join in, the given region will contain great number of information for the agents to make their choices. And each agent’s choice contributes to the ultimate aggregation, though not with equal contribution. Consequently, the final results of entrepreneurial ratios will be consistently converged to some fixed points. Our result partly confirms the theory provided by Andersson and Larsson [56], who emphasized the importance of entrepreneurial neighbors on the individuals’ decisions to become entrepreneurs. The practical implication of this conclusion is that for the region where entrepreneurial agglomeration is advocated, it should contain a certain number of agents (the potential entrepreneurs or workers) at its early age. In another word, the population of labors for the region is the basis for entrepreneurial agglomeration formation, especially in the early age. 

(3) The role model in the region on the formation of entrepreneurial agglomeration to Minniti, et al. [68], the imitation of entrepreneurial activity across different markets encourages more individuals to become entrepreneurs. Our model theoretically proved that the new agents will imitate the agents who have become incumbent entrepreneurs, and the role models of entrepreneurship are very essential to the formation of entrepreneurial agglomeration, but they will present a decreasing capability in stimulating entrepreneurship. This conclusion basically supports the theory by Zheng and Du [38] and Minniti, Bygrave and practice [68]. While in our study, the function of role model is a little different from theirs. Our assumptions are that the individuals will evaluate the ER and then decide whether they can accept the risks of entrepreneurship in the region. A role model is more successful in creating new jobs and starting new firms, thus he can provide more information and spillover more knowledge to the surroundings. When new entrants can observe more role models and get more information about entrepreneurship for a given region, their risk-taking abilities will accordingly be improved and their sense to start new businesses will be encouraged. However, as shown in our simulation, the role models will present a decreasing capability in stimulating entrepreneurship. The possible explanation could be that when the number of agents becomes huge, the small number of role models cannot compete with the whole situation, as shown in section 5.1; furthermore, the role model will dominate the market and absorb more employment, which could threaten the new entrepreneurs and potentially reduce the entrepreneurial activities. From practical perspective, this conclusion implies that the introducing more role models early is more effective in concentrating the entrepreneurial activities.

(4) The initial ER in the region on the formation of entrepreneurial agglomeration

Through mathematical analysis and numerical simulation we have basically proved that when the initial ER in a region is higher, the probability to converge at a high level will increase (as shown in Fig. 9). This theoretical conclusion is conformed to the bandwagon effect on the entrepreneurial choice [62,63,69], which implies that the entrepreneurial history of the region is important. When a region is with higher initial ER, the individuals will face high probability to meet the entrepreneurs, and they will also be easily absorbed in to take similar career. Under such situation, the region will exhibit the character of path dependency. Another reason is that high initial ER will clear away more ambiguity for entrepreneurship, with which the new entrants can be more inclined to this kind of career. This conclusion partially confirmed the “social advantage” theories [55-57] and “economic advantage” theory [51], but our conclusion emphasizes the increasing-return process to adoption, and this process is dynamic and fluctuating before it comes to the final agglomeration, which could be a new finding to our knowledge. For the reason that the initial ER in the region is more important on the formation of entrepreneurial agglomeration that later ER, the practical meaning is that introducing more entrepreneurial activities or increasing the initial ER when the region is at small scale or in early stage is more effective in forming the agglomeration.”

2. In section 5 “Conclusion and Suggestion” (see page 26-27), we add the summarized theoretical and practical implications to our study.

“Our findings have some theoretical implications: it is proved that entrepreneurship has “memory” and the entrepreneurial history could have influence on the future, which can be repressed as another kind of bandwagon effect; it is found that 4 determinants from agents and regions can contribute to the formation of entrepreneurial agglomeration, thus focusing on improving individual’s risk-taking abilities, increasing the initial population scale and entrepreneurial ratio in the region, and introducing more entrepreneur role models in the early age can improve the formation of entrepreneurial agglomeration.

Also, our findings have some practical implication. Considering the strong effects from “memory” and “history” in entrepreneurship, our policy should be more concentrated on the determinants from “early” age, such as attracting more role models and introducing more entrepreneurial activities when the region is at small scale. Besides, due to the great contribution on self-reinforcement, lowering the risk compensation value for individuals is very important. The policy point can focus on improving entrepreneurial education, enhancing entrepreneurial experience so as to improve the individuals’ risk-taking abilities and creating fair businesses environment, supporting more resources to the individuals, or nurturing more business opportunities to downsize the risks in the regions.”

Besides, we have made some minor corrections and modifications on the paper’s logic, consistency, completeness, and persuasiveness, etc.. All corrections and modifications are marked in red.

Responses to Reviewer-2

Reviewer #2: This is an interesting paper and I enjoyed reading it. However, there are essential weaknesses that need to be addressed.

1) The introductory/opening section should communicate a little clearer the literature gaps, as well as the study's aims & objectives in order to facilitate the flow of the study.

Response: First of all, we would like to thank reviewer for such valuable suggestions. Your suggestions really helped us to improve this paper. We have modified the introductory/opening section, and make it more clearly in aims and objectives, and well as the literature gaps. The modified texts are shown below (see page 3-5).

“As prior studies proved, the new career chooser will definitely make his decision by many criterion, and some internal factors such as his advantages of experience [4,5], attitudes [6,7], risk-taking capabilities [8,9], the propensity in the career type [10,11], his expectation in the future career [12,13] etc., or some external factors such as regional opportunities [14], favorable environment [15], or the social status [16], etc.. Though the final career choice is determined by both internal and external factors, prior research have mostly assumed that the individual is well informed before his decision, no matter by his own endeavor or through his social connections or on new business model [17-19].

However, to our knowledge, the existing study may be weak on answering two questions on entrepreneurial agglomeration formation: the first one is why there are different levels of entrepreneurial agglomerations in different regions, even in the regions with similar economic and social conditions. According to our observation, this different level of agglomeration enlightens us to assume that entrepreneurship has “memory” and the entrepreneurial history could have influence on the future. However, prior studies haven’t given answers to it. The second is that prior studies have set their research on too strict assumptions, especially when the individual’s capabilities on obtaining and utilizing information are considered. As mentioned before, prior literatures have mostly assumed that the individual is well informed before his entrepreneurial decision. However, for most individuals, the region’s social circumstances, such as prospect of employment, business environment, education, or the supporting methods, could be barriers to their perception and not unanimously understood or obtained due to exchange cost or other reasons [20]. That is to say, for most people, they cannot wholly understand the situations for the given region. Under this event, the individuals will refer to some easily obtained information as the criteria on career choice.

The level of entrepreneurial agglomeration can be expressed as entrepreneurial ratio for any given region [21], and it can be easily obtained when the region’s business facilities or local people are observed [22]. In order to discover the mechanism of entrepreneurial agglomeration formation, we will focus more on the index of entrepreneurial ratio. And, since the entrepreneurial ratio is determined by individual career choices between entrepreneurship and employment, our point will be transferred to the dynamic of micro-decisions of career choices on entrepreneurial agglomeration formation. Our purpose is to provide a theoretical framework describing the dynamics of entrepreneurial agglomeration formation and answer 3 questions: what are the factors that determine the entrepreneurial agglomeration? why dose entrepreneurship mostly prosper in some regions while not in other regions?, and why the region with similar situations cannot replicate the entrepreneurial agglomeration as others?

Unlike traditional research which focused on individual’s attributes or on the environmental factors that affect the individual’s choice on entrepreneurial career, our assumption is that the new entrant determined to make his career choice cannot fully obtain or understand the region’s circumstances. Alternatively, some information or resources, such as the proximity to incumbent entrepreneurs, could be easily obtained or understood. Specifically, if a new entrant can observe the existing ratio of entrepreneurship in the region, his perception about the likelihood to join this career will be reinforced. That is to say, the latter one will refer to the entrepreneurial ratio produced by the precedent’s involvement, which is kind of social selection. We assume all the entrants are heterogeneous, and their reactions vary when faced with different level of entrepreneurial activities. Based on social selection and after repeated choices sequentially, the process will converge to some fixed points, and this region will exhibit the landscape of variation when entrepreneurship ratio is concerned. Furthermore, different from prior study that mostly adopt traditional empirical method, we use mathematical method and numerical simulation to depict the repeated entering processes and reveal the dynamics of the level of existing entrepreneurial activity on the formation of entrepreneurial agglomeration. We think the mathematical analysis can better reflect the mechanism of career choices for individuals and explain the determinants for entrepreneurial agglomeration formation, and the numerical simulation can better depict the whole formation process step by step, which are always regarded as “black box” for traditional empirical studies [23].

This study makes two major contributions. First, we advocate that the formation of entrepreneurial agglomeration is a process of social selection by individuals. Specifically, we found that the career choice of individuals is affected by social environment, especially by the observable conditions such as the scale of existing entrepreneurship. And after repeated selection by the individuals, the agglomeration will emerge following the rule of Polya process. Not like most studies that confess the new agents will make their decision on entrepreneurship based on clear information, out study insists that the individuals can make their choices based on very basic information, and the existing entrepreneurship will have a self-reinforcing capability to absorb or expel the new entrant. This finding confirms that entrepreneurship has “memory” and the entrepreneurial history could have influence on the future. Second, along with our study, we confirm that some determinants from social selection can affect the formation of entrepreneurial agglomeration. In detail, our results indicate that the agent’s risk compensation value, the initial scale of agents, the number of entrepreneur role models from the region’s early stage and the initial entrepreneurial ratio in the region can be involved to contribute the formation of entrepreneurial agglomeration. Thus exerting influence on the determinants from an early age, and improving the individual’s risk-taking abilities are beneficial to the formation of entrepreneurial agglomeration.”

2) Overall there are good arguments and well researched points made in this paper, but I feel that author needs to take to a further level. I recommend that you include the following references focused on the target journal and on the paper’s topics:

Al-Mamary, Y. H., & Alshallaqi, M. (2022). Impact of autonomy, innovativeness, risk-taking, proactiveness, and competitive aggressiveness on students’ intention to start a new venture. Journal of Innovation & Knowledge, 7(4), 100239. https://10.1016/j.jik.2022.100239

Bouncken, R. B., Lapidus, A., & Qui, Y. (2022). Organizational sustainability identity:‘New Work’of home offices and coworking spaces as facilitators. Sustainable Technology and Entrepreneurship, 1(2), 100011. https://doi.org/10.1016/j.stae.2022.100011

Dana, L. P., Salamzadeh, A., Hadizadeh, M., Heydari, G., & Shamsoddin, S. (2022). Urban Entrepreneurship and Sustainable Businesses in Smart Cities: Exploring the Role of Digital Technologies. Sustainable Technology and Entrepreneurship, 100016. 1 (2) https://doi.org/10.1016/j.stae.2022.100016

Dijkstra, H., van Beukering, P., & Brouwer, R. (2022). Marine plastic entrepreneurship; Exploring drivers, barriers and value creation in the blue economy. Sustainable Technology and Entrepreneurship, 1(3), 100018. https://doi.org/10.1016/j.stae.2022.100018

Guo, H., Guo, A., & Ma, H. (2022). Inside the black box: How business model innovation contributes to digital start-up performance. Journal of Innovation & Knowledge, 7(2), 100188. https://https://doi.org/10.1016/j.jik.2022.100188

Huang, Y., Li, P., Wang, J., & Li, K. (2022). Innovativeness and entrepreneurial performance of female entrepreneurs. Journal of Innovation & Knowledge, 7(4), 100257. https://10.1016/j.jik.2022.100257

Méndez-Picazo, M., Galindo-Martín, M., & Castaño-Martínez, M. (2021). Effects of sociocultural and economic factors on social entrepreneurship and sustainable development. Journal of Innovation & Knowledge, 6(2), 69-77. https://10.1016/j.jik.2020.06.001

Metallo, C., Agrifoglio, R., Briganti, P., Mercurio, L., & Ferrara, M. (2021). Entrepreneurial Behaviour and New Venture Creation: the Psychoanalytic Perspective. Journal of Innovation & Knowledge, 6(1), 35-42. https://10.1016/j.jik.2020.02.001

Response: thank you for your suggestion. After careful check, we think some literatures are very relevant and useful, so some literatures such as “Méndez-Picazo, M., Galindo-Martín, M., & Castaño-Martínez, M. (2021). Effects of sociocultural and economic factors on social entrepreneurship and sustainable development. Journal of Innovation & Knowledge, 6(2), 69-77.”” Guo, H., Guo, A., & Ma, H. (2022). Inside the black box: How business model innovation contributes to digital start-up performance. Journal of Innovation & Knowledge, 7(2), 100188.” are included in our paper.

3) The research is well-developed.

Response: thank you.

4) At the end of, the author should include clear statements as to where research should now go.

Response: as your comment, we have added this part to our paper, also, the limitation of our research is supplemented (see 5. “Conclusion and Suggestion”, page 27-28, marked in red font).

“One limitation of our study is confined to theoretical analysis, because robustness, validation and accuracy of our conclusions are not verified empirically. Our study adopts the model proceeded by mathematical analysis and numerical simulation, which can clearly exhibit the formation process of entrepreneurial agglomeration and conclude the determinants under strict model construction conditions, but have some weakness in introducing the real data to verify whether our conclusions can sufficiently reflect the real situations. Another limitation is that the determinants from our study are only confined to 4 categories; therefore, some other variables from prior research cannot be verified in our model. Our model has strong theoretical foundation, and most adopted determinants are suggested or mentioned by prior study, thus some of our findings also confirm to prior studies. However, limited by the mathematical model, we only adopt some of the variables; whether other variables could be regarded as the dynamics has not been proofed in our analysis.

Future research should modify the mathematical model, making it more acceptable to the variables suggested by prior research and observed from real situations, and more capable in explaining the mechanism of entrepreneurial agglomeration formation. In addition, empirical analysis combining with our theoretical model and using real data from various regions is also recommended in the future study.”

5) Carefully check the references, so as to make sure they are all complete and follow the Guidelines to Authors.

Response: we have checked them, and all of them are complete and follow the Guidelines to Authors.

6) Finally, when you submit the corrected version, please do check thoroughly, in order to avoid grammar, syntax or structure/presentation flaws.

Response: we have checked the paper thoroughly.

Besides, we have made some minor corrections and modifications on the paper’s logic, consistency, completeness, and persuasiveness, etc.. All corrections and modifications are marked in red font.

---

## [Decision Letter · Decision Letter 1]

13 Jun 2023

PONE-D-22-27945R1The Dynamics of Entrepreneurial Agglomeration Formation: Social Selection and SimulationPLOS ONE

Dear Dr. Javeed,

Thank you for submitting your manuscript to PLOS ONE. After careful consideration, we feel that it has merit but does not fully meet PLOS ONE’s publication criteria as it currently stands. Therefore, we invite you to submit a revised version of the manuscript that addresses the points raised during the review process.

We look forward to receiving your revised manuscript.

Kind regards,

Valentina Diana Rusu, PhD

Academic Editor

PLOS ONE

Journal Requirements:

Reviewers' comments:

Reviewer's Responses to Questions

**Comments to the Author**

1. If the authors have adequately addressed your comments raised in a previous round of review and you feel that this manuscript is now acceptable for publication, you may indicate that here to bypass the “Comments to the Author” section, enter your conflict of interest statement in the “Confidential to Editor” section, and submit your "Accept" recommendation.

Reviewer #1: (No Response)

Reviewer #2: All comments have been addressed

2. Is the manuscript technically sound, and do the data support the conclusions?

Reviewer #1: Partly

Reviewer #2: Yes

3. Has the statistical analysis been performed appropriately and rigorously? 

Reviewer #1: I Don't Know

Reviewer #2: Yes

4. Have the authors made all data underlying the findings in their manuscript fully available?

Reviewer #1: Yes

Reviewer #2: Yes

5. Is the manuscript presented in an intelligible fashion and written in standard English?

Reviewer #1: Yes

Reviewer #2: Yes

6. Review Comments to the Author

Reviewer #1: This paper is related to an interesting topic, namely The Dynamics of Entrepreneurial Agglomeration Formation: Social Selection and Simulation for PLOS ONE

However, the manuscript needs to be further developed in order to meet the expected academic requirements.

1) Separate Conclusion from Suggestion

Add limitations and direction for future research in new section

2) Theoretical and practical implications is missing. Provide a more critical discussion regarding theoretical and practical implications. What are the theoretical and practical implications that can be derived from this study?

• Add practical implications

• Theoretical Implications

Reviewer #2: Nothing

7. PLOS authors have the option to publish the peer review history of their article (what does this mean?). If published, this will include your full peer review and any attached files.

Reviewer #1: No

Reviewer #2: No

---

## [Author Response · Author response to Decision Letter 1]

27 Jun 2023

There are two reviewers who have checked our paper. Reviewer 1 provides some suggestions on our modification, and the Review Comments to the Author from reviewer 2 is “nothing”. Therefore, our responses to the comments are mainly focused on reviewer 1.

Reviewer #1: This paper is related to an interesting topic, namely The Dynamics of Entrepreneurial Agglomeration Formation: Social Selection and Simulation for PLOS ONE. However, the manuscript needs to be further developed in order to meet the expected academic requirements.

Response: Dear reviewer, we highly appreciate your efforts for reviewing our paper and giving valuable suggestions. It really helped us to improve the quality of paper.

 1) Separate Conclusion from Suggestion

Add limitations and direction for future research in new section.

Response: Thank you for the comments. We have separated Conclusion from Suggestion, and added Limitations and Direction for Future Research in new section (under Section 5, page 25-28, marked in red)

5. Conclusion and Suggestion

5.1 Conclusion

The entrepreneurial agglomeration, as an effective means to sustain regional economic development, has been widely emphasized globally, especially in emerging countries such as China. However, numerous cases show that the fostering and consolidating entrepreneurial agglomeration is not an easy task. For lots of regions with similar social and economic circumstances, the ultimate results for the entrepreneurial agglomeration vary dramatically though the local governments have paid great endeavors in forging them. Then we are facing the problems of “what are the factors that determine the entrepreneurial agglomeration”, “why entrepreneurship mostly prospers in some regions while not in other regions?” and “why the region with similar situations cannot replicate the entrepreneurial agglomeration as others?”

In order to explore the determinants of entrepreneurial agglomeration formation, traditional literatures have mostly sought evidences from the individuals or the regional conditions. However, as our observation, the individuals cannot clearly obtain and understand all necessary information and resources for entrepreneurship in the given region. That is to say, traditional information for the new entrants is generally ambiguous. Along with this judgement, the individuals will seek for much easier criterion to make their career choices. Entrepreneurial agglomeration is reflected as the aggregated entrepreneurship. For any region, the new entrant is fully exposed to the environment where he is with some probability to meet the entrepreneurs, and the bandwagon effect will drive him much easily affected by his proximity to take similar activity. Under such argument, we provide a theoretical framework describing the dynamics of entrepreneurial agglomeration formation. Unlike the traditional research that assume the individuals are informed all the necessary information in the given region, we introduce the hypothesis that the new entrants will make their career choices (entrepreneurship or employment) based on existing entrepreneurial ratio. According to our model, after repeated choices, this trajectory conforms to a nonlinear Polya process, i.e., the whole process will emerge the self-reinforcing and path-dependency characters. Then after numerical calculation and simulation, we confirmed that the process under our assumption will converge to some fixed points, which can exhibit the formation of entrepreneurial agglomeration and explain why different regions will vary in their levels of entrepreneurial agglomeration.

Our main findings are concentrated on the discovery of the dynamic process of and the determinants for the formation of entrepreneurial agglomeration. Under our assumption, our model can explain the formation of entrepreneurial agglomeration is a nonlinear Polya process, and the repeated entrants’ career choices will lead to the convergence of entrepreneurship in any given region. As to the determinants, our analysis and simulation have proved at least four categories can be included: the agent’s risk compensation value, the initial scale of agents, the number of role models (especially in the early stage), and the initial entrepreneurial ratio. Accordingly, they can basically answer three questions in section 1.

5.2 Implications

Our findings have some theoretical implications: it is proved that entrepreneurship has “memory” and the entrepreneurial history could have influence on the future, which can be expressed as another kind of bandwagon effect emerging from entrepreneurial activities; it is concluded theoretically that 4 determinants from agents and regions, i.e., the agent’s risk compensation value, the initial scale of agents, the number of role models (especially in the early stage), and the initial entrepreneurial ratio, can contribute to the formation of entrepreneurial agglomeration, thus focusing on improving individual’s risk-taking abilities, increasing the initial population scale and entrepreneurial ratio in the region, and introducing more entrepreneur role models in the early age can improve the formation of entrepreneurial agglomeration.

Also, our findings have some practical implications. Considering the strong effects from “memory” and “history” in entrepreneurship, our policy should be more concentrated on the determinants from “early” age, such as attracting more role models and introducing more entrepreneurial activities when the region is at its initiation stage or in small scale. Besides, due to the great contribution on self-reinforcement, lowering the risk compensation value for individuals is very important. The policy point can focus on improving entrepreneurial education, enhancing entrepreneurial experience so as to improve the individuals’ risk-taking abilities and creating fair businesses environment, supporting more resources to the individuals, or nurturing more business opportunities to downsize the risks in the regions.

5.3 Limitations and Direction for Future Research

One limitation of our study is confined to theoretical analysis, because robustness, validation and accuracy of our conclusions are not verified empirically. Our study adopts the model proceeded by mathematical analysis and numerical simulation, which can clearly exhibit the formation process of entrepreneurial agglomeration and conclude the determinants under strict model construction conditions, but have some weakness in introducing the real data to verify whether our conclusions can sufficiently reflect the real situations. Another limitation is that the determinants from our study are only confined to 4 categories; therefore, some other variables from prior research cannot be verified in our model. Our model has strong theoretical foundation, and most adopted determinants are suggested or mentioned by prior study, thus some of our findings also confirm to prior studies. However, limited by the mathematical model, we only adopt some of the variables; whether other variables could be regarded as the dynamics has not been proofed in our analysis.

Future research should modify the mathematical model, making it more acceptable to the variables suggested by prior research and observed from real situations, and more capable in explaining the mechanism of entrepreneurial agglomeration formation. In addition, empirical analysis combining with our theoretical model and using real data from various regions is also recommended in the future study.

2) Theoretical and practical implications is missing. Provide a more critical discussion regarding theoretical and practical implications. What are the theoretical and practical implications that can be derived from this study?

• Add practical implications

• Theoretical Implications

Response: Thank you for the comment. We have refined the theoretical and practical implications and added them to our manuscript, all of which are marked in red color (under Section 5, page 27).

5.2 Implications

Our findings have some theoretical implications: it is proved that entrepreneurship has “memory” and the entrepreneurial history could have influence on the future, which can be expressed as another kind of bandwagon effect emerging from entrepreneurial activities; it is concluded theoretically that 4 determinants from agents and regions, i.e., the agent’s risk compensation value, the initial scale of agents, the number of role models (especially in the early stage), and the initial entrepreneurial ratio, can contribute to the formation of entrepreneurial agglomeration, thus focusing on improving individual’s risk-taking abilities, increasing the initial population scale and entrepreneurial ratio in the region, and introducing more entrepreneur role models in the early age can improve the formation of entrepreneurial agglomeration.

Also, our findings have some practical implications. Considering the strong effects from “memory” and “history” in entrepreneurship, our policy should be more concentrated on the determinants from “early” age, such as attracting more role models and introducing more entrepreneurial activities when the region is at its initiation stage or in small scale. Besides, due to the great contribution on self-reinforcement, lowering the risk compensation value for individuals is very important. The policy point can focus on improving entrepreneurial education, enhancing entrepreneurial experience so as to improve the individuals’ risk-taking abilities and creating fair businesses environment, supporting more resources to the individuals, or nurturing more business opportunities to downsize the risks in the regions.

At the end, we must thank you again. We hope this version of our paper would satisfy you.

---

## [Decision Letter · Decision Letter 2]

24 Jul 2023

PONE-D-22-27945R2The Dynamics of Entrepreneurial Agglomeration Formation: Social Selection and SimulationPLOS ONE

Dear Dr. Javeed,

Thank you for submitting your manuscript to PLOS ONE. After careful consideration, we feel that it has merit but does not fully meet PLOS ONE’s publication criteria as it currently stands. Therefore, we invite you to submit a revised version of the manuscript that addresses the points raised during the review process.

We look forward to receiving your revised manuscript.

Kind regards,

Valentina Diana Rusu, PhD

Academic Editor

PLOS ONE

Journal Requirements:

Reviewers' comments:

Reviewer's Responses to Questions

**Comments to the Author**

1. If the authors have adequately addressed your comments raised in a previous round of review and you feel that this manuscript is now acceptable for publication, you may indicate that here to bypass the “Comments to the Author” section, enter your conflict of interest statement in the “Confidential to Editor” section, and submit your "Accept" recommendation.

Reviewer #1: (No Response)

2. Is the manuscript technically sound, and do the data support the conclusions?

Reviewer #1: Yes

3. Has the statistical analysis been performed appropriately and rigorously? 

Reviewer #1: Yes

4. Have the authors made all data underlying the findings in their manuscript fully available?

Reviewer #1: Yes

5. Is the manuscript presented in an intelligible fashion and written in standard English?

Reviewer #1: Yes

6. Review Comments to the Author

Reviewer #1: Provide a more critical discussion regarding theoretical and practical implications. What are the theoretical and practical implications that can be derived from this study?

• Add practical implications

• Theoretical Implications

7. PLOS authors have the option to publish the peer review history of their article (what does this mean?). If published, this will include your full peer review and any attached files.

Reviewer #1: No

---

## [Author Response · Author response to Decision Letter 2]

28 Jul 2023

Reviewer #1: Provide a more critical discussion regarding theoretical and practical implications. What are the theoretical and practical implications that can be derived from this study?

• Add practical implications

• Theoretical Implications

Response: Dear reviewer, first of all, we would like thank the reviewer for giving valuable suggestions throughout the peer review process. Your comments really helped us to further improve the quality of the paper. We have refined the theoretical and practical implications and added them to our manuscript, all of which are marked in red color.

“5.2 Implications

Practical implications

Firstly, considering the strong effects from “memory” and “history” in entrepreneurship, the policy should be more concentrated on the determinants from “early” age, such as attracting more role models and introducing more entrepreneurial activities when the region is at its initiation stage or in small scale. In detail, the government can start with the evaluation of a region’s entrepreneurial activities and make up polices to gather more role entrepreneurs and potential entrepreneurs into the region by cutting off taxes, providing more subsidies, or issuing more favorable measures to start and operate the businesses.

Secondly, due to the great contribution on self-reinforcement effect, lowering the risk compensation value for individuals is very important. The policy point can focus on fostering the traits on entrepreneurship. In detail, the government should invest more on improving entrepreneurship education, encouraging more potential entrepreneurs to observe, witness or participate in the incumbent business to enhance their entrepreneurial experience, so as to improve the individuals’ risk-taking abilities

Finally, for the reason that the improvement of entrepreneurial infrastructure is a good means to lower the entrepreneurship risks, and accordingly reduce the risk compensation value for the potential entrepreneurs. The policy can exert its ability on creating a fairer businesses environment, building more entrepreneurial facilities such as incubators, transportation and communication systems etc., to nurture more business opportunities and facilitate more entrepreneurial activities.

Theoretical implications

Firstly, we explore the dynamics of entrepreneurial formation. It is proved that the formation of entrepreneurial agglomeration is a self-reinforcing process. Based on social selection by individuals, we confirm that the formation of entrepreneurial agglomeration can be described as a Polya process, and the existing entrepreneurship will absorb or expel the new entrant. In our theoretical framework, the entrepreneurship has “memory” and the entrepreneurial history could have influence on the future, which can be expressed as another kind of bandwagon effect emerging from entrepreneurial activities.

Secondly, we explore the determinants which contribute to the entrepreneurial formation. Based on mathematical analysis and simulation, we conclude that some determinants from social selection can affect the formation of entrepreneurial agglomeration. In detail, our results theoretically indicate that 4 determinants from agents and regions, i.e., the agent’s risk compensation value, the initial scale of agents, the number of role models (especially in the early stage), and the initial entrepreneurial ratio, can contribute to the formation of entrepreneurial agglomeration, thus focusing on improving individual’s risk-taking abilities, increasing the initial population scale and entrepreneurial ratio in the region, and introducing more entrepreneur role models in the early age can improve the formation of entrepreneurial agglomeration.”

Thank you for your comments!

---

## [Decision Letter · Decision Letter 3]

4 Sep 2023

The Dynamics of Entrepreneurial Agglomeration Formation: Social Selection and Simulation

PONE-D-22-27945R3

Dear Dr. Javeed,

We’re pleased to inform you that your manuscript has been judged scientifically suitable for publication and will be formally accepted for publication once it meets all outstanding technical requirements.

Kind regards,

Valentina Diana Rusu, PhD

Academic Editor

PLOS ONE

Additional Editor Comments (optional):

Reviewers' comments:

Reviewer's Responses to Questions

**Comments to the Author**

1. If the authors have adequately addressed your comments raised in a previous round of review and you feel that this manuscript is now acceptable for publication, you may indicate that here to bypass the “Comments to the Author” section, enter your conflict of interest statement in the “Confidential to Editor” section, and submit your "Accept" recommendation.

Reviewer #1: All comments have been addressed

2. Is the manuscript technically sound, and do the data support the conclusions?

Reviewer #1: Yes

3. Has the statistical analysis been performed appropriately and rigorously? 

Reviewer #1: I Don't Know

4. Have the authors made all data underlying the findings in their manuscript fully available?

Reviewer #1: Yes

5. Is the manuscript presented in an intelligible fashion and written in standard English?

Reviewer #1: Yes

6. Review Comments to the Author

Reviewer #1: (No Response)

7. PLOS authors have the option to publish the peer review history of their article (what does this mean?). If published, this will include your full peer review and any attached files.

Reviewer #1: No

---

## [Editor Report · Acceptance letter]

11 Sep 2023

PONE-D-22-27945R3 

The Dynamics of Entrepreneurial Agglomeration Formation:
Social Selection and Simulation 

Dear Dr. Javeed:

I'm pleased to inform you that your manuscript has been deemed suitable for publication in PLOS ONE. Congratulations! Your manuscript is now with our production department. 

Kind regards, 

on behalf of

Dr. Valentina Diana Rusu 

Academic Editor

PLOS ONE